# Disentanglement of Variations with Multimodal Generative Modeling

**Yijie Zhang, Yiyang Shen, Weiran Wang**
Department of Computer Science
University of Iowa
Iowa City, IA 52242, USA
{yijie-zhang,yiyang-shen,weiran-wang}@uiowa.edu

## Abstract

Multimodal data are prevalent across various domains, and learning robust representations of such data is paramount to enhancing generation quality and downstream task performance. To handle heterogeneity and interconnections among different modalities, recent multimodal generative models extract shared and private (modality-specific) information with two separate variables. Despite attempts to enforce disentanglement between these two variables, these methods struggle with challenging datasets where the likelihood model is insufficient. In this paper, we propose **I**nformation-**D**isentangled **M**ultimodal **VAE** (IDMVAE) to explicitly address this issue, with rigorous mutual information-based regularizations, including cross-view mutual information maximization for extracting shared variables, and a cycle-consistency style loss for redundancy removal using generative augmentations. We further introduce diffusion models to improve the capacity of latent priors. These newly proposed components are complementary to each other. Compared to existing approaches, IDMVAE shows a clean separation between shared and private information, demonstrating superior generation quality and semantic coherence on challenging datasets.

## 1 Introduction

Most real-world data are inherently multimodal or multi-view[1]. Videos contain both visual scenes and sounds (Zhao et al., 2018; Owens & Efros, 2018; Chen et al., 2020a; Gong et al., 2023; Kim et al., 2024); robots can see and feel via sensors (Lee et al., 2019); images are often accompanied by captions (Radford et al., 2021; Jia et al., 2021); and heterogeneous human, animal, and environmental data are collected for health improvements (Adisasmito et al., 2022). In addition to these naturally occurring data, synthetic multi-view data constructed from semantically similar input components or via augmentation are also widely used to learn useful representations for downstream tasks (Veličković et al., 2019; Chen et al., 2020b; Caron et al., 2020; Tian et al., 2020a; Bardes et al., 2022). Despite the abundance of such data, leveraging them is nontrivial even with naturally aligned modalities due to their diversity and complex correlations. Therefore, a core challenge is to integrate information across views to learn universal, transferable representations.

Variational autoencoders (VAEs, Kingma & Welling, 2014) and their multimodal extensions have emerged as a powerful paradigm to tackle this problem (Wang et al., 2016; Suzuki et al., 2016). They can extract useful shared information in data with missing modalities (Wu & Goodman, 2018) and noise (Shi et al., 2021). While early works have assumed that a single latent space can capture all relevant information and data variations (Shi et al., 2019; Sutter et al., 2021), recent approaches have recognized the existence of both shared and modality-specific (private) information in real-world datasets (Daunhawer et al., 2022; Lee & Pavlovic, 2021; Palumbo et al., 2023; 2024). However, modeling shared and private components naturally exposes a challenge:

> *How can we achieve maximal disentanglement between shared and private variables so that learned representations are complete and non-redundant?*

---

[1]We use "modality" and "view" interchangeably as they both appear in the literature.

Without a clean separation, shared information leaks into private encodings and vice versa, causing weak coherence across modalities, wasted model capacity, and inadequate generation quality.

**Our Contributions.** We propose Information-Disentangled Multimodal VAE (IDMVAE), a novel framework for unsupervised multimodal representation learning hailing from theoretical stringency and practical performance. (1) Different from prior works which use capacity-based (Wang et al., 2016) or shortcut-preventing (Palumbo et al., 2023) heuristics, we employ mutual information (MI)-based regularizations to ensure disentanglement. In particular, we use cross-view MI to extract common factors that likelihood models fail to fully capture, thereby enhancing cross-modal coherence. Additionally, a cycle-consistency-style loss removes redundancy between shared and private latents using samples generated by the model itself, eliminating the need for domain-specific augmentations. (2) To overcome limitations of simple Gaussian priors, we leverage diffusion-based priors (Sohl-Dickstein et al., 2015; Ho et al., 2020; Song et al., 2021) that capture the richness of multimodal latent spaces, leading to greater representational capacities. (3) Across multiple complex datasets spanning image, text, and multi-omics data, IDMVAE performs consistently better than state-of-the-art methods in terms of cross-modal generation and coherence, showing a synergy between MI-based objectives and diffusion priors, which leads to improved performance.

## 2 METHOD

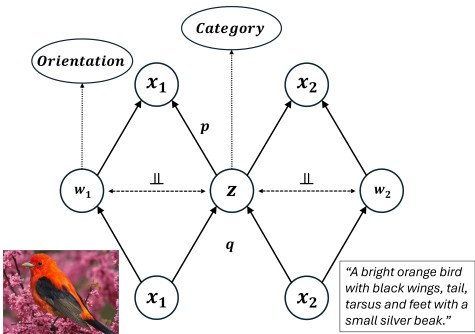

Figure 1: Our approach with two modalities.

Given a set of $M$ modalities $\mathbf{X} = \{\mathbf{x}_1, \ldots, \mathbf{x}_M\}$, we would like to learn a factorized latent space for each modality $m$, separating information shared across modalities, captured by latent variable $\mathbf{z}$, from modality-specific private information, captured by $\mathbf{w}_m$. As illustrated in Figure 1, we assume that, $\forall m$, $\mathbf{x}_m$ is generated by $\mathbf{z}_m$ and $\mathbf{w}_m$ jointly with $p(\mathbf{x}_m|\mathbf{z}_m, \mathbf{w}_m)$, and the latent variables have independent priors, i.e., $\mathbf{z} \perp\!\!\!\perp \mathbf{w}_m$, and $p(\mathbf{z}, \{\mathbf{w}_m\}_{m=1}^M) = p(\mathbf{z}) \prod_{m=1}^M p(\mathbf{w}_m)$. We perform variational inference and parameterize the approximate posteriors with a factorized form: $q(\mathbf{z}, \mathbf{w}_m|\mathbf{x}_m) = q(\mathbf{z}|\mathbf{x}_m) \cdot q(\mathbf{w}_m|\mathbf{x}_m)$. Learned $q(\mathbf{z}|\mathbf{x}_m)$ and $q(\mathbf{w}_m|\mathbf{x}_m)$ provide representations of original inputs which can be used in downstream tasks. Our method consists of three components.

### 2.1 LIKELIHOOD MODELING WITH MULTIMODAL VAE

We begin with the objective of the MMVAE+ model (Palumbo et al., 2023) as our foundation. Let $q(\mathbf{z}|\mathbf{x}_m)$ be the posterior of shared variable derived from modality $m$ (with distribution parameters modeled by an encoder), and $q(\mathbf{w}_m|\mathbf{x}_m)$ be the posterior of private variable from modality $m$ (modeled by another encoder). One can define a global posterior of $\mathbf{z}$ by aggregating view-specific posteriors, e.g., using the mixture-of-experts (MoE) scheme $q(\mathbf{z}|\mathbf{X}) = \frac{1}{M} \sum_{m=1}^M q(\mathbf{z}|\mathbf{x}_m)$. Together with generative distributions $p(\mathbf{x}_m|\mathbf{z}, \mathbf{w}_m)$ modeled by decoders of each view, an Evidence Lower Bound (ELBO) for $\mathbf{X}$ can be derived with variational inference (Kingma & Welling, 2014), which involves reconstructing $\mathbf{x}_n$ using samples of $q(\mathbf{w}_n|\mathbf{x}_n)$ and $q(\mathbf{z}|\mathbf{x}_m)$, for $n, m = 1, \ldots, M$.

Since the posterior $q(\mathbf{w}_n|\mathbf{x}_n)$ is dependent on $\mathbf{x}_n$ and potentially retains shared information, to keep decoder $p(\mathbf{x}_n|\mathbf{z}, \mathbf{w}_n)$ from taking the "shortcut" to use leaked shared information, MMVAE+ uses sample of $\tilde{\mathbf{w}}_n$ from an auxiliary prior $r(\tilde{\mathbf{w}}_n)$ instead for cross-view reconstruction, where $\mathbf{z}$ is derived from another view $m \neq n$; avoiding such shortcut enforces shared information to come from $q(\mathbf{z}|\mathbf{x}_m)$. This design leads to the following minimizing objective:

$$\mathcal{L}_{\text{MMVAE+}} = -\frac{1}{M} \sum_{m=1}^M \mathbb{E}_{\substack{\mathbf{z}\sim q(\mathbf{z}|\mathbf{x}_m) \\ \mathbf{w}_m\sim q(\mathbf{w}_m|\mathbf{x}_m) \\ \{\tilde{\mathbf{w}}_n\sim r(\tilde{\mathbf{w}}_n)\}_{n\neq m}}} \left[ \log \left( \frac{p(\mathbf{x}_m|\mathbf{z}, \mathbf{w}_m)p(\mathbf{z})p(\mathbf{w}_m)}{q(\mathbf{z}|\mathbf{X})q(\mathbf{w}_m|\mathbf{x}_m)} \prod_{n\neq m} p(\mathbf{x}_n|\mathbf{z}, \tilde{\mathbf{w}}_n) \right) \right].$$

Within the expectation, $p(\mathbf{x}_m|\mathbf{z}, \mathbf{w}_m)$ and $p(\mathbf{x}_n|\mathbf{z}, \tilde{\mathbf{w}}_n)$ are conditional likelihood of self- and cross-reconstructions, respectively. Palumbo et al. (2023) shows that $-\mathcal{L}_{\text{MMVAE+}}(\mathbf{x}_{1:M}) \leq \log(\mathbf{x}_{1:M})$

remains a valid ELBO. Compared with previous multimodal VAE using the MoE parameterization (Shi et al., 2019), MMVAE+ achieves superior performance for extracting shared information, although the result is somewhat sensitive to the relative capacity (dimensionality) of $\mathbf{z}$ and $\mathbf{w}$. In the rest of this section, we will show that regularizing the generative model with mutual information (MI) is more effective at achieving disentanglement. We emphasize that MMVAE+ is one option for multimodal VAE, and our improvement below can also be applied to other models, e.g., PoE (Wu & Goodman, 2018) or MoPoE (Sutter et al., 2021).

## 2.2 Shared Variable Extraction with Cross-view MI Maximization

While general (per-dimension) disentanglement of variations is theoretically challenging (Locatello et al., 2019), the underlying structure of our setup that inputs of different modalities share a common cause facilitates (variable-level) disentanglement of shared versus private information.

To extract the shared information, it is natural to enforce the shared representation of modality $m$, denoted $\mathbf{z}_m$ (with distribution $q(\mathbf{z}|\mathbf{x}_m)$) to have high mutual information (MI) with $\mathbf{x}_n$ for $n \neq m$. We note this is partially pursued by $\mathcal{L}_{\text{MMVAE+}}$, indirectly through the cross-view reconstruction: in light of the decomposition $I(\mathbf{z}_m, \mathbf{w}_n; \mathbf{x}_n) = I(\mathbf{z}_m; \mathbf{x}_n) + I(\mathbf{w}_n; \mathbf{x}_n|\mathbf{z}_m)$, we can maximize $I(\mathbf{z}_m; \mathbf{x}_n)$ by maximizing $I(\mathbf{z}_m, \mathbf{w}_n; \mathbf{x}_n)$ while minimizing $I(\mathbf{w}_n; \mathbf{x}_n|\mathbf{z}_m)$. Focusing on $I(\mathbf{z}_m, \mathbf{w}_n; \mathbf{x}_n) = H(\mathbf{x}_n) - H(\mathbf{x}_n|\mathbf{z}_m, \mathbf{w}_n)$ where the entropy $H(\mathbf{x}_n)$ is a constant, minimizing the conditional entropy $H(\mathbf{x}_n|\mathbf{z}_m, \mathbf{w}_n) = \mathbb{E}_{\mathbf{x}_n, \mathbf{z}_m, \mathbf{w}_n}[-\log p(\mathbf{x}_n|\mathbf{z}_m, \mathbf{w}_n)]$ is equivalent to maximizing conditional likelihood, as done by MMVAE+. However, maximizing this upper bound $I(\mathbf{z}_m, \mathbf{w}_n; \mathbf{x}_n)$ alone does not ensure maximal $I(\mathbf{z}_m; \mathbf{x}_n)$, due to the gap $I(\mathbf{w}_n; \mathbf{x}_n|\mathbf{z}_m)$. Therefore, likelihood maximization alone does not ensure the extraction of a shared variable.

We thus take the alternative approach to maximize $I(\mathbf{z}_m; \mathbf{z}_n)$ which is a lower bound of $I(\mathbf{z}_m; \mathbf{x}_n)$: $I(\mathbf{z}_m; \mathbf{x}_n) = I(\mathbf{z}_m; \mathbf{z}_n, \mathbf{x}_n) - I(\mathbf{z}_m; \mathbf{z}_n|\mathbf{x}_n) = I(\mathbf{z}_m; \mathbf{z}_n, \mathbf{x}_n) = I(\mathbf{z}_m; \mathbf{z}_n) + I(\mathbf{z}_m; \mathbf{x}_n|\mathbf{z}_n) \geq I(\mathbf{z}_m; \mathbf{z}_n)$, where $I(\mathbf{z}_m; \mathbf{z}_n|\mathbf{x}_n) = 0$ in the first step due to variability of $\mathbf{z}_n$ coming from $\mathbf{x}_n$ only (Federici et al., 2020). In this work, we use the contrastive estimate of MI (Oord et al., 2018):

$$I(\mathbf{z}_m; \mathbf{z}_n) \approx Contrast(\mathbf{z}_m, \mathbf{z}_n) := \mathbb{E}_{\mathbf{z}_m, \mathbf{z}_n} \log \left[ \frac{\phi(\mathbf{z}_m, \mathbf{z}_n)}{\phi(\mathbf{z}_m, \mathbf{z}_n) + \sum_{j=1}^{k} \phi(\mathbf{z}_m, \bar{\mathbf{z}}_n^j)} \right] \quad (1)$$

where $\phi(\mathbf{z}_m, \mathbf{z}_n) = \exp\left(\frac{\mathbf{z}_m^\top \mathbf{z}_n}{\|\mathbf{z}_m\| \cdot \|\mathbf{z}_n\|}\right)$ is the affinity function, and $\{\bar{\mathbf{z}}_n^j\}_{j=1}^{k}$ are $k$ negative examples randomly sampled from the minibatch not aligned with $\mathbf{z}_m$. Since we have $M$ modalities, we compute the average of cross-modality MIs as our regularization for extracting shared information:

$$\mathcal{L}_{\text{CrossMI}} = -\frac{2}{M(M-1)} \sum_{m<n} Contrast(\mathbf{z}_m, \mathbf{z}_n).$$

## 2.3 Disentanglement with Generative Augmentation

While equation 1 encourages $\mathbf{z}$ to capture shared information across views, it does not guarantee that the learned $\mathbf{z}_m$ contains no private information which should be modeled by $\mathbf{w}_m$. Similarly, even if $\mathbf{z}_m$ contains no private information and the self-reconstruction term in $\mathcal{L}_{\text{MMVAE+}}$ encourages $(\mathbf{z}_m, \mathbf{w}_m)$ to jointly capture all information about $\mathbf{x}_m$, the learned $\mathbf{w}_m$ can still retain shared information. Thus, we need additional regularization to remove redundancy between $\mathbf{z}_m$ and $\mathbf{w}_m$.

To motivate our method, consider the desired scenario where $\mathbf{z}_m$ and $\mathbf{w}_m$ are disentangled, so that they each can be varied independently to generate new samples of $\mathbf{x}_m$ using the decoder. Let $\mathbf{x}_m$ and $\mathbf{x}'_m$ be two input samples, and let $(\mathbf{z}_m, \mathbf{w}_m)$ be a pair of samples drawn from the posteriors $q(\mathbf{z}|\mathbf{x}_m)$ and $q(\mathbf{w}_m|\mathbf{x}_m)$, respectively, and similarly $(\mathbf{z}'_m, \mathbf{w}'_m)$ be a pair of samples drawn from conditional posteriors for $\mathbf{x}'_m$. With disentanglement and a good likelihood model, a sample $\mathbf{x}_m^+ \sim p(\mathbf{x}_m|\mathbf{z}_m, \mathbf{w}'_m)$ would share the same $\mathbf{z}$ with $\mathbf{x}_m$. In turn, when we map $\mathbf{x}_m^+$ back to the latent space, $q(\mathbf{z}|\mathbf{x}_m^+)$ and $q(\mathbf{z}|\mathbf{x}_m)$ should be similar. Likewise, $q(\mathbf{w}_m|\mathbf{x}_m^+)$ and $q(\mathbf{w}_m|\mathbf{x}'_m)$ should be similar.

More formally, assume that $\mathbf{z}_m$ is sufficient for $\mathbf{x}_n$, meaning that it captures all shared information, i.e., $I(\mathbf{z}_m; \mathbf{x}_n) = I(\mathbf{x}_m; \mathbf{x}_n)$ as encouraged by $\mathcal{L}_{\text{CrossMI}}$. Then in view of $I(\mathbf{z}_m; \mathbf{x}_n) = H(\mathbf{z}_m) - H(\mathbf{z}_m|\mathbf{x}_n)$, we would like to find the minimal $\mathbf{z}_m$ (with lowest $H(\mathbf{z}_m)$) by minimizing

$$H(\mathbf{z}_m|\mathbf{x}_n) = \mathbb{E}_{\mathbf{z}_m, \mathbf{x}_n}[-\log p(\mathbf{z}_m|\mathbf{x}_n)] \approx \mathbb{E}_{\mathbf{X} \sim p(\mathbf{X}), \mathbf{z}_m \sim q(\mathbf{z}|\mathbf{x}_m)}[-\log q(\mathbf{z} = \mathbf{z}_m|\mathbf{x}_n)].$$

Similar approaches have been used by Federici et al. (2020, symmetric KL for minimizing $I(\mathbf{x}_m, \mathbf{z}_m | \mathbf{x}_n)$) and Tsai et al. (2019, inverse prediction) for learning minimally sufficient shared variable. Essentially, for extracting shared information, $\mathbf{x}_m$ and $\mathbf{x}_n$ indeed constitute two views that are mutually redundant, satisfying $I(\mathbf{x}_m; \mathbf{x}_n | \mathbf{z}) = 0$, so that the IB principle naturally applies.

Note, however, we do not have multiple natural views sharing $\mathbf{w}_m$ to carry out the above idea. This challenge motivates us to synthesize the view $\mathbf{x}_m^+ \sim p(\mathbf{x}_m | \mathbf{z}_m, \mathbf{w}_m')$, with $\mathbf{w}_m' \sim q(\mathbf{w} | \mathbf{x}_m')$ from another sample $\mathbf{x}_m'$, to reduce the redundancy of learned $\mathbf{w}_m$ by approximately minimizing $H(\mathbf{w}_m | \mathbf{x}_m')$ with the following loss

$$\mathbb{E}_{\mathbf{x}_m \sim p(\mathbf{x}_m), \mathbf{x}_m' \sim p(\mathbf{x}_m), \mathbf{z}_m \sim q(\mathbf{z} | \mathbf{x}_m), \mathbf{w}_m' \sim q(\mathbf{w}_m | \mathbf{x}_m'), \mathbf{x}_m^+ \sim p(\mathbf{x}_m | \mathbf{z}_m, \mathbf{w}_m')}[-\log q(\mathbf{w}_m' | \mathbf{x}_m^+)].$$

Assuming that posteriors are parameterized Gaussians, this loss reduces to $\ell_2$ loss for matching means of posteriors. In practice, we find it more effective to use a contrastive loss for matching, i.e.,

$$\mathcal{L}_{\text{GenAug},\mathbf{w}_m} := -Contrast(\mathbf{w}_m'', \mathbf{w}_m') \qquad \text{where} \quad \mathbf{x}_m^+ \sim p(\mathbf{x}_m | \mathbf{z}_m, \mathbf{w}_m'), \ \mathbf{w}_m'' \sim q(\mathbf{w}_m | \mathbf{x}_m^+).$$

We show results obtained with contrastive estimation in the main paper, and provide empirical comparisons of the two implementations in Appendix A. We also define $\mathcal{L}_{\text{GenAug},\mathbf{z}_m}$ similarly by switching the role of $\mathbf{z}_m$ and $\mathbf{w}_m$. The total redundancy removal regularization is defined as

$$\mathcal{L}_{\text{GenAug}} = \frac{1}{2M} \sum\nolimits_{m=1}^{M} \left( \mathcal{L}_{\text{GenAug},\mathbf{z}_m} + \mathcal{L}_{\text{GenAug},\mathbf{w}_m} \right).$$

Although we do not have the reconstruction target for $\mathbf{x}_m^+$, matching $q(\mathbf{z} | \mathbf{x}_m^+)$ with $q(\mathbf{z} | \mathbf{x}_m)$, and matching $q(\mathbf{w}_m | \mathbf{x}_m^+)$ with $q(\mathbf{w}_m | \mathbf{x}_m')$ implement a form of *cycle-consistency* (Zhu et al., 2017), and provide learning signals for both the encoder and the decoder. Previously, Bai et al. (2021) derived an ELBO of sequence data for disentangling static versus dynamic components, which involved mutual information terms based on data augmentation, similar to $\mathcal{L}_{\text{GenAug}}$. However, their augmentation requires strong domain knowledge (e.g., shuffling the frame order does not alter the static component, and color change applied to all frames does not alter the dynamic component). In contrast, our augmentations require no domain knowledge and are produced by the model itself.

## 2.4 THE FINAL IDMVAE OBJECTIVE

We define our objective of Information-Disentangled Multimodal VAE (IDMVAE) as

$$\min \ \mathcal{L}_{\text{IDMVAE}} := \mathcal{L}_{\text{MMVAE+}} + \lambda_1 \mathcal{L}_{\text{CrossMI}} + \lambda_2 \mathcal{L}_{\text{GenAug}} \tag{2}$$

where $\lambda_1$ and $\lambda_2$ are user parameters tuned on the validation set.

**Diffusion Priors** In most multimodal VAEs, the prior distributions are chosen to be simple and easy to sample from, e.g., Gaussian for continuous data. However, such unstructured priors may not be ideal for representation learning, whose purpose is to discover the useful structure of data for supervised downstream tasks. As an example, a representation containing rich label information most likely have a clustering structure where data of different classes are separated far apart, and will not have a uni-modal distribution like the Gaussian. We use diffusion models (Sohl-Dickstein et al., 2015; Ho et al., 2020; Song et al., 2021) to overcome this limitation by parameterizing $p(\mathbf{z})$ as a denoising process started with pure noise. To naturally introduce diffusion models into our loss, we decompose the KL divergence inside $\mathcal{L}_{\text{MMVAE+}}$ (which we minimize) as (Vahdat et al., 2021):

$$D_{\text{KL}}(q(\mathbf{z} | \mathbf{x}) || p(\mathbf{z})) = \mathbb{E}_{q(\mathbf{z} | \mathbf{x})} \left[ \log q(\mathbf{z} | \mathbf{x}) \right] + \mathbb{E}_{q(\mathbf{z} | \mathbf{x})} \left[ -\log p(\mathbf{z}) \right]. \tag{3}$$

The first term maximizes the entropy of the approximate posterior $q(\mathbf{z} | \mathbf{x})$. The second term maximizes the likelihood of samples from $q(\mathbf{z} | \mathbf{x})$ under $p(\mathbf{z})$, which we model with diffusion models. We can treat $\mathbf{z} \sim q(\mathbf{z} | \mathbf{x})$ as "data", and destroy its structure by gradually adding noise to it, resulting in pure noise after a number of steps. With repeated applications of a denoising network, diffusion models gradually reverse the noising process and recover the original data from pure noise. Diffusion models have well-defined ELBO objectives which *lower bound* $\log p(\mathbf{z})$, and plugging them into equation 3 yields *valid upper bounds* of the KL divergence. Since the latent variables are of low dimensionality, we parameterize the diffusion backward process with a simple feedforward network. In practice, we introduce additional loss weight for $\mathbb{E}_{q(\mathbf{z} | \mathbf{x})} \left[ -\log p(\mathbf{z}) \right]$, and model the mean of $q(\mathbf{z} | \mathbf{x})$ with the DDPM parameterization (Ho et al., 2020). We optimize over all modules (encoders, decoders, diffusion networks) jointly in an *end-to-end* manner. A recent work (Palumbo et al., 2024) proposed a two-step approach which first learns the representations with MMVAE+, and then learns diffusion models in the input space, conditioned on VAE reconstructions. Note our use of diffusion model has a different motivation, and we jointly train it during representation learning.

## 3 RELATED WORKS

**Disentanglement in VAEs.** To achieve disentangled latent representations in VAEs, researchers have commonly used mutual information (MI) based regularization, and employed various metrics to assess the results (Higgins et al., 2017; Kim & Mnih, 2018; Chen et al., 2018; Kumar et al., 2018). However, it has been shown that, without supervision or inductive bias in the model, it is theoretically challenging to recover (per-dimension) disentanglement (Locatello et al., 2019).

**Contrastive and self-supervised learning (SSL).** SSL is applied to a single modality, with artificial views created based on the structures of data (Oord et al., 2018; Logeswaran & Lee, 2018; Hjelm et al., 2019; Bachman et al., 2019; Chen et al., 2020b; Caron et al., 2020; Tian et al., 2020a; Bardes et al., 2022; Zbontar et al., 2021), as well as multimodal data (Radford et al., 2021; Jia et al., 2021; Elizalde et al., 2023), and many methods are motivated by the classical infomax principle (Linsker, 1988) and they implement neural estimation of mutual information, with contrastive loss being the most popular variant. Recent works have proposed theoretical interpretations of SSL and contrastive learning (Wang & Isola, 2020; Zimmermann et al., 2021; Hyvärinen et al., 2019; Tian et al., 2020b; Tosh et al., 2021; Chen et al., 2021; Zhai et al., 2024), with the focus of providing guarantees for extracting the shared variable, without considering the private variables.

A few works took private variations into consideration. von Kügelgen et al. (2021) proposed a generative model in which the latent space is divided into "content" and "style"; importantly, data augmentations were assumed to preserve content while altering dimensions within style. Tsai et al. (2021) studied self-supervised learning from a multi-view perspective and with the *multi-view redundancy* assumption (Chaudhuri et al., 2009; Tosh et al., 2021) that the private variable of each view contains little information for the downstream task, they focused on extracting the shared variable with combinations of several multi-view losses. Realizing the limitation of this assumption, Liang et al. (2023) studied the scenario where the private variables contain significant useful information, and proposed a contrastive learning algorithm for extracting it. Their algorithm required sophisticated data augmentation procedures designed for the downstream task. Lyu et al. (2022) proposed a model for understanding SSL, assuming a data generation process similar to ours. They extracted the shared variable with CCA loss and the private variable by MI minimization.

**Information bottleneck (IB) and mutual-information regularization.** Another set of probabilistic models was motivated by the IB method (Tishby et al., 1999; Tishby & Zaslavsky, 2015; Achille & Soatto, 2018). Alemi et al. (2017) proposed a variational IB method to extract $\mathbf{z}$ from $\mathbf{x}_1$ which has high MI with $\mathbf{x}_2$ (estimated with conditional likelihood), so that it captures the shared information, and at the same time has low MI with $\mathbf{x}_1$ so that it contains little nuisance factors/private information. Federici et al. (2020) leveraged the multi-view redundancy assumption that all the information $\mathbf{x}_1$ contains about an unobserved label is also contained in $\mathbf{x}_2$, and showed that that if the learned representation $\mathbf{z}$ is sufficient, in the sense that $I(\mathbf{x}_1, \mathbf{x}_2|\mathbf{z}) = 0$, then $\mathbf{z}$ has all the predictive power from $(\mathbf{x}_1, \mathbf{x}_2)$ for label. Remarkably, their objective did not involve any reconstruction paths, and the authors considered this to be an advantage, given that density modeling for high dimensional data is difficult. Wang et al. (2025) extended Federici et al. (2020) and proposed a two-step approach to first extract shared and then private variables with guarantees, again without generative modeling. We argue that, with the development of powerful generative models, likelihood modeling becomes feasible and provides the additional benefit of (controllable) generation.

**Multimodal latent diffusions.** Recent works have proposed to first learn independent VAEs for each modality, and then couple their latent representation with diffusion models in a second training stage (Bounoua et al., 2023; Wesego & Rooshenas, 2024). A key limitation of this approach is that separately trained VAEs may fail to capture cross-modal correlations, leading to a loss of information that the diffusion model cannot subsequently recover. In contrast, our latent prior is jointly trained with the rest of the objective and regularization terms, with the gradient of the latent diffusion loss propagated to the encoders. More importantly, these works do not perform disentanglement of the shared versus private variables, and therefore do not offer the same level of control as we do, e.g., we can combine and from different data points or samples from the prior to generate new inputs.

## 4 EXPERIMENTS

We compare our method, IDMVAE, and its variant with diffusion priors, against several baselines.

Table 1: Latent classification on PolyMNIST-Quadrant. Accuracies are averaged over 5 modalities. For methods with a single latent variable in each view, evaluation results are collected under $z$.

| Model | $z \rightarrow Digit \uparrow$ | $z \rightarrow Quad \downarrow$ | $w \rightarrow Quad \uparrow$ | $w \rightarrow Digit \downarrow$ |
|---|---|---|---|---|
| MMVAE | 0.492 | 0.798 | — | — |
| MoPoE-VAE | 0.536 | 0.751 | — | — |
| DMVAE | 0.157 | **0.254** | 0.710 | 0.179 |
| MMVAE+ | 0.382 | 0.355 | 0.999 | 0.341 |
| SBM-VAE | 0.263 | 0.995 | — | — |
| IDMVAE (ours) | **0.983** | 0.271 | 0.999 | 0.162 |
| $\quad - \mathcal{L}_{\text{CrossMI}} (\lambda_1 = 0)$ | 0.111 | 0.267 | 0.999 | 0.356 |
| $\quad - \mathcal{L}_{\text{GenAug}} (\lambda_2 = 0)$ | 0.977 | 0.277 | 0.999 | 0.202 |
| + Diffusion prior | 0.982 | 0.267 | 0.999 | **0.143** |

**MMVAE** (Shi et al., 2019): uses a MoE inference network to combine information from different modalities. It only models the shared variable **z** with ELBO.

**MoPoE-VAE** (Sutter et al., 2021): uses a mixture-of-products-of-experts inference network for **z**.

**DMVAE** (Lee & Pavlovic, 2021): performs PoE inference for **z**, and models $\mathbf{w}_m$ within ELBO.

**MMVAE+** (Palumbo et al., 2023): performs separation of shared versus private information with the help of auxiliary prior variables. It is a special case of IDMVAE (w.o. diffusion) with $\lambda_1 = \lambda_2 = 0$.

**DisentangledSSL** (Wang et al., 2025): performs extraction of shared variable (using the method of Federici et al., 2020) and private variable in two sequential steps. It is a state-of-the-art disentanglement method without likelihood modeling, but it can only be applied to two views currently.

**SBM-VAE** (Wesego & Rooshenas, 2024): first trains individual VAEs for each modality with a single latent variable, and then couples modalities with diffusion modeling on the joint representations.

## 4.1 RESULTS ON POLYMNIST-QUADRANT

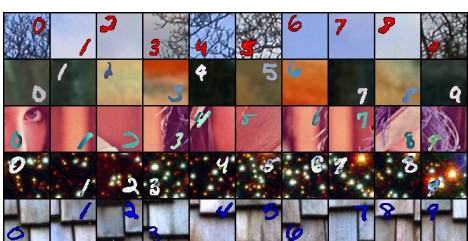

Figure 2: PolyMNIST-Quadrant dataset. Digits (0-9) are placed in one of the four quadrants randomly. Each column contains one multimodal sample. Each modality has a different background scheme. The digit label is shared across all modalities, while quadrant label is private to each modality.

PolyMNIST (Sutter et al., 2021) is a benchmark for multimodal representation learning, consisting of MNIST (LeCun et al., 1998) digits overlaid on complex backgrounds. We make the dataset more challenging by taking each MNIST digit and placing a 32x32 scaled version of it into one of four quadrants of a 64x64 canvas; see Figure 2 for an illustration. This modification introduces the private latent variable which captures the quadrant position (with ground truth label) and background for each modality, allowing for nuanced evaluation of disentanglement and generation. Our training/validation/test sets contain 220,000/5,000/10,000 samples. We use the deep residual network (He et al., 2015) architecture as the backbone of encoders and decoders for all methods. The dimensionality is set to 32 for **z** and 128 for $\mathbf{w}_m$.

**Latent Classification.** For evaluation, we perform linear classification on the *samples* of posterior distributions (samples reflect both mean and variance of posteriors). Multi-class logistic regression models are trained on the posterior samples of the training set and applied to the posterior samples of the test set. We perform two types of classifications: (1) predicting shared label from the shared variable **z** and private label from the private variable $\mathbf{w}_m$, where high accuracy is better, indicating the desired variation is captured; and (2) cross-classification, where we predict shared label from $\mathbf{w}_m$ and predict private label from **z**. Ideally, with successful disentanglement, cross-classification accuracies should approach the performance of a random classifier (e.g., 10% for predicting digits from $\mathbf{w}_m$, 25% for predicting quadrants from **z**). We present results of different methods in Table 1, as well as performance of our method when either $\mathcal{L}_{\text{CrossMI}}$ or $\mathcal{L}_{\text{GenAug}}$ is removed from our loss. Clearly, our method achieves superior performance. $\mathcal{L}_{\text{CrossMI}}$ is critical for extracting the shared

Table 2: Generative coherence, averaged over 5 views, on PolyMNIST-Quadrant. We use subscript $q$ to indicate samples from posteriors and subscript $p$ to indicate samples from priors. For generated images, digit label is determined by $\mathbf{z}_{s,q}$ or otherwise random (with target accuracy 10%), quadrant label is determined by $\mathbf{w}_{s,q}$ or otherwise random (with target accuracy 25%).

| Model | Self Gen ($s = t$) | | | | Cross Gen ($s \neq t$) | Uncond. |
| | $Gen(\mathbf{z}_{s,q}, \mathbf{w}_{t,p})$ | | $Gen(\mathbf{z}_{t,p}, \mathbf{w}_{s,q})$ | | $Gen(\mathbf{z}_{s,q}, \mathbf{w}_{t,p})$ | $Gen(\mathbf{z}_p, \mathbf{w}_p)$ |
| | $Digit \uparrow$ | $Quad \downarrow$ | $Digit \downarrow$ | $Quad \uparrow$ | $Digit \uparrow$ | $Digit \uparrow$ |
|---|---|---|---|---|---|---|
| MMVAE | — | — | — | — | 0.170 | 0.041 |
| MoPoE-VAE | — | — | — | — | 0.173 | 0.029 |
| DMVAE | 0.297 | 0.252 | 0.532 | 0.999 | 0.161 | 0.005 |
| MMVAE+ | 0.120 | 0.251 | 0.915 | 0.999 | 0.119 | 0.000 |
| SBM-VAE | — | — | — | — | 0.158 | 0.007 |
| IDMVAE (ours) | 0.898 | 0.249 | 0.162 | 0.999 | 0.881 | 0.070 |
| $- \mathcal{L}_{\text{CrossMI}}$ ($\lambda_1 = 0$) | 0.101 | 0.252 | 0.926 | 0.999 | 0.100 | 0.000 |
| $- \mathcal{L}_{\text{GenAug}}$ ($\lambda_2 = 0$) | 0.670 | 0.250 | 0.370 | 0.999 | 0.671 | 0.008 |
| + Diffusion prior | **0.942** | 0.251 | **0.106** | 0.999 | **0.887** | **0.664** |

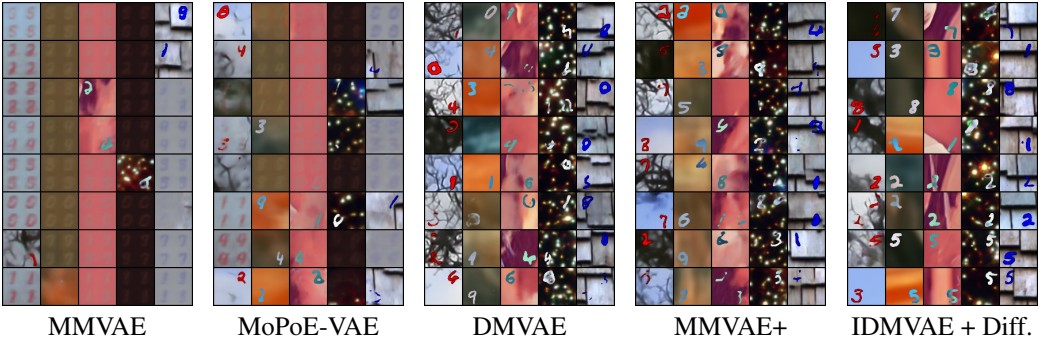

|  |  |  |  |  |
|---|---|---|---|---|
| MMVAE | MoPoE-VAE | DMVAE | MMVAE+ | IDMVAE + Diff. |

Figure 3: Unconditional generations on PolyMNIST-Quadrant. Each row is a multimodal sample generated with a prior sample of $\mathbf{z}$, so images in the same row ideally have the same digit identity.

variable, and this is because the digits occupy a small number of pixels and pure likelihood modeling may ignore them. $\mathcal{L}_{\text{GenAug}}$ helps remove redundant information, so that cross-classification accuracy is reduced. Adding diffusion in latent space (last row of table) leads to small gain.

**Conditional Coherence.** This metric evaluates the model's ability to generate consistent samples across modalities. We assess this for both self-reconstruction and cross-modal generation. Formally, we combine either posterior $\mathbf{z}_{s,q} \sim q(\mathbf{z}|\mathbf{x}_s)$ or prior $\mathbf{z}_{s,p} \sim p(\mathbf{z})$ (using diffusion prior if available) of a modality $s$, with the posterior $\mathbf{w}_{t,q} \sim q(\mathbf{w}|\mathbf{x}_t)$ or the prior $\mathbf{w}_{t,p} \sim p(\mathbf{w}_t)$ (using diffusion prior if available) of modality $t$, and apply $p(\mathbf{x}_t|\mathbf{z}, \mathbf{w}_t)$ to generate a new sample of modality $t$. This sample should have the same digit label as $\mathbf{x}_s$ if the posterior of $\mathbf{z}$ is used, and a random digit label if the prior is used. Similarly, the quadrant label can be determined based on whether posterior or prior is used for $\mathbf{w}_t$. We then use ResNet classifiers trained on original images to predict corresponding labels of generated images, and the averaged accuracy across modalities is referred to as *coherence*. We provide conditional generative coherence in Table 2 (left panel for self generation where $s = t$, and middle panel for cross generation $s \neq t$); see Appendix B.2 for sample generations. The results are consistent with those of latent classification, and diffusion priors significantly boost coherence.

**Unconditional Coherence.** This metric further assesses the consistency of the shared information in unconditionally generated samples. We first sample a shared latent code $\mathbf{z}_p \sim p(\mathbf{z})$ (using diffusion prior when available). For each modality $m$, we then sample an independent private $\mathbf{w}_{m,p} \sim p(\mathbf{w}_m)$ and generate a sample $\hat{\mathbf{x}}_m$ from the combined latent code $(\mathbf{z}_p, \mathbf{w}_{m,p})$. The generated multimodal sample $\{\hat{x}_1, ..., \hat{x}_M\}$ are then passed to their respective digit classifiers (ResNet) trained on original training images, to predict the shared label. A sample set is considered coherent if *all* classifiers agree on the same shared label. We report the percentage of coherent sets as unconditional coherence, shown in Table 2 (right panel). Most methods obtain close to zero unconditional coherence, indicating the difficulty of matching prior and posterior distributions for latent variables.

Table 3: Latent classification on CUB-HQ, using posterior means. $\mathbf{z}_1$ and $\mathbf{w}_1$ refers to image latents.

| Model | $\mathbf{z} \to Cat. \uparrow$ | $\mathbf{z}_1 \to Dir. \downarrow$ | $\mathbf{w}_1 \to Dir. \uparrow$ | $\mathbf{w} \to Cat. \downarrow$ |
|---|---|---|---|---|
| MMVAE | 0.602 | 0.727 | — | — |
| MoPoE-VAE | 0.615 | 0.777 | — | — |
| DMVAE | 0.511 | 0.658 | 0.732 | 0.394 |
| MMVAE+ | 0.633 | 0.787 | 0.567 | 0.486 |
| SBM-VAE | 0.452 | 0.799 | — | — |
| DisentangledSSL | 0.672 | 0.538 | 0.567 | **0.173** |
| IDMVAE (ours) | 0.764 | 0.592 | 0.677 | 0.322 |
| $- \mathcal{L}_{\text{CrossMI}}$ ($\lambda_1 = 0$) | 0.614 | 0.792 | 0.557 | 0.379 |
| $- \mathcal{L}_{\text{GenAug}}$ ($\lambda_2 = 0$) | **0.777** | **0.498** | 0.792 | 0.395 |
| + Diffusion prior | 0.752 | 0.526 | **0.797** | 0.311 |

However, with diffusion prior, our method achieves significantly better coherence, thanks to its flexibility. We show generations in Figure 3, and 2D visualizations of latent codes in Appendix B.3.

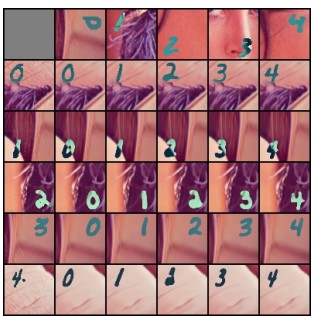

Figure 4: Augmentations.

**Generative augmentation.** Recall that in $\mathcal{L}_{\text{GenAug}}$ we mix and match posteriors of $\mathbf{z}$ and $\mathbf{w}_m$ from different samples to generate new samples in modality $m$. We provide illustration of such samples from our trained model in Figure 4. The first row and first column contain images for which we extract posterior samples of $\mathbf{z}$ and $\mathbf{w}_m$ respectively. And the rest of the grid contain generate images using samples of $\mathbf{w}_m$ of the corresponding row and $\mathbf{z}$ of the corresponding column. We observe that images in each column share the same digit, while images in each row share the same quadrant, as desired. Generated images are of high quality, showing that we can independently vary shared and private variables to obtain controllable generations.

## 4.2 RESULTS ON HIGH-QUALITY CUB (CUB-HQ)

The CUB-200-2011 dataset (Wah et al., 2011; Reed et al., 2016; Shi et al., 2019; Palumbo et al., 2023; 2024) is a widely used benchmark for fine-grained visual categorization, containing RGB images of 200 bird species. Each image is paired with 10 textual descriptions. Following Palumbo et al. (2024), we group 22 categories of species from the 200 bird species into 8 super-categories, yielding 1-of-8 class labels for these species. Data with category label is split into training/validation/test with 70%/15%/15% portions. The rest 178 species are added to the training set for representation learning. The training/validation/test sets contain 113,900/1,940/2,040 samples, respectively.

In this experiment, we use the 256x256 resolution version of the original CUB dataset (Wah et al., 2011) *without* cropping images using bounding boxes for birds. Compared to the cropped version used in prior works (Shi et al., 2019; Palumbo et al., 2023), in our setup, each image contains richer background and is more challenging for representation learning. For this dataset, the two modalities (image, text) share rich information about bird category, since the text describes the color of different parts of the bird. To evaluate the quality of private information, we note that the horizontal direction of the bird (with direction inferred from the original CUB attributes) can only be inferred from the image. Therefore, we consider the direction as a private label for the image modality. We refer to this dataset as high-quality CUB (CUB-HQ); see more details on data generation in Appendix C.1.

We pre-process the image modality with a pretrained-VAE encoder[2] similarly used by Peebles & Xie (2022), converting each original RGB images to a $4 \times 32 \times 32$ tensor, which then serves as the model input. We use ResNet as encoders and decoders for pre-processed input images, while convolution network as those for texts (using one-hot representation of text with a vocabulary of 1,590 words). The dimensionality is set to 256 for both $\mathbf{z}$ and $\mathbf{w}_m$.

**Representation quality** We perform linear classification on the latent representations $\mathbf{z}$ and $\mathbf{w}$ to exam their information content; the results are given in Table 3. Note that with disentangled latent representations, the target classification accuracy is 50% (i.e., random) for predicting direction from

---

[2]https://huggingface.co/stabilityai/sd-vae-ft-mse

Table 4: Generative coherence, as measured by FID and CLIPscores on the CUB-HQ test set. As a reference, the CLIPScore between ground truth test images and text is 0.762. For image-to-image generation, we use the posterior of $\mathbf{z}$ and the prior of $\mathbf{w}$.

| Model | Text-to-Img | | Img-to-text | Img-to-Img | |
|---|---|---|---|---|---|
| | **FID** ↓ | **CLIP** ↑ | **CLIP** ↑ | **FID** ↓ | **CLIP** ↑ |
| DMVAE | 104.167 | 0.665 | 0.683 | 70.534 | 0.707 |
| MMVAE+ | 70.157 | 0.691 | 0.693 | 62.528 | 0.712 |
| SBM-VAE | 79.900 | 0.684 | 0.687 | — | — |
| IDMVAE (ours) | 64.435 | 0.718 | 0.736 | **58.065** | **0.721** |
| $-\mathcal{L}_{\text{CrossMI}}$ ($\lambda_1 = 0$) | 72.166 | 0.694 | 0.692 | 62.938 | 0.710 |
| $-\mathcal{L}_{\text{GenAug}}$ ($\lambda_2 = 0$) | 66.291 | 0.709 | 0.719 | 69.988 | 0.702 |
| + Diffusion prior | **60.549** | **0.721** | **0.737** | 59.700 | 0.716 |

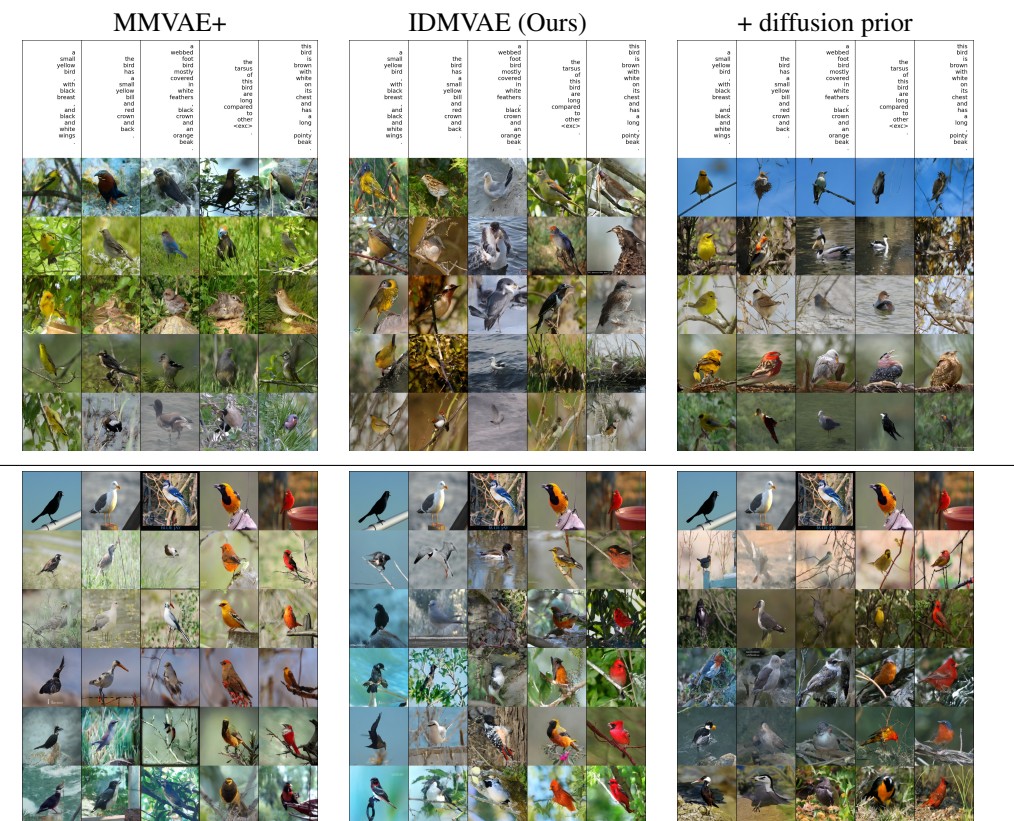

Figure 5: **Top**: Text-to-image generation on CUB-HQ, with the DiT denoiser. We combine the posterior sample $\mathbf{z}$ of the text (top row) with the prior sample $\mathbf{w}$ of the image modality (shared by each row) for generation. Ideally, images in the same column are of similar bird category and color, and images in the same row have the same bird orientation. **Bottom**: Image-to-image generation. We combine the posterior sample $\mathbf{z}$ of the image (top row) with the prior sample $\mathbf{w}$ of the image modality (shared by each row) for generation. Ideally, images in the same column are of similar bird category and color, while images in the same row have the same bird orientation.

$\mathbf{z}$, and 12.5% for predicting category from $\mathbf{w}_1$ (derived from image). We observe that cross-view MI is critical for recovering $\mathbf{z}$, when we do not have a very strong likelihood model (due to limited image data). On the other hand, generative augmentation significantly reduces redundancy in the latent space by removing the directional information from $\mathbf{z}$ and capturing it in $\mathbf{w}$. Adding diffusion prior leads to the best performance overall, demonstrating the effectiveness of a flexible latent prior. Our methods outperform both generative and deterministic baselines by a clear margin.

**Generative coherence** After training, our model can generate images of 4 channels and $32 \times 32$ resolution, which can then be decoded using the pretrained-VAE decoder into $256 \times 256$ RGB

Table 5: Prediction accuracy on TCGA, averaged over 2 modalities and 5 splits.

| Model | $\mathbf{z}\uparrow$ | $\mathbf{w}$ | $\mathbf{z}+\mathbf{w}\uparrow$ |
|---|---|---|---|
| MMVAE | 0.695±0.010 | — | — |
| MoPoE-VAE | 0.695±0.014 | — | — |
| DMVAE | 0.688±0.018 | 0.691±0.014 | 0.697±0.016 |
| MMVAE+ | 0.692±0.010 | 0.690±0.012 | 0.690±0.011 |
| DisentangledSSL | 0.691±0.011 | 0.691±0.012 | 0.690±0.011 |
| IDMVAE (ours) | 0.707±0.016 | 0.708±0.013 | 0.718±0.017 |
| $-\mathcal{L}_{\text{CrossMI}}$ ($\lambda_1=0$) | 0.691±0.014 | 0.689±0.010 | 0.691±0.014 |
| $-\mathcal{L}_{\text{GenAug}}$ ($\lambda_2=0$) | 0.701±0.015 | 0.706±0.019 | 0.723±0.013 |
| + Diffusion prior | **0.714±0.009** | **0.719±0.024** | **0.731±0.019** |

images. However, given the relatively small training set size, the generated images tend to be blurry. Following Pandey et al. (2022) and Palumbo et al. (2024), we train a diffusion model to generate high-quality images conditioned on our model's generations, using a DiT architecture (Peebles & Xie, 2022). This DiT acts as a denoiser for our model; see Appendix C.2 for more implementation details. To measure the generative coherence of models, we perform image-to-text, text-to-image, and image-to-image generations. We measure the quality of generated high-resolution images with FID (Heusel et al., 2017), and the coherence between text and images using CLIPScore (Hessel et al., 2021). These metrics are given in Table 4, and our model's generation performance exceeds the baselines. We provide samples of text-to-image and image-to-image generations in Figure 5.

In Appendix C.3, we provide more generations with and without the DiT denoiser. The success of our approach shows that $\mathcal{L}_{\text{GenAug}}$ is effective even when the generative augmentations are not of very high quality. While the IDMVAE samples produced without DiT exhibit blurriness, they successfully capture core features (color and orientation of the bird), thereby facilitating disentanglement. The DiT denoiser, finetuned separately, can then fill in the details for high-quality generation.

## 4.3 RESULTS ON THE CANCER GENOME ATLAS (TCGA)

TCGA dataset[3] is a real-world multi-omics dataset that is by nature multimodal. Using the same data processing procedure from Lee & van der Schaar (2021), we obtain a dataset of 10,960 samples (of which 9,477 are labeled) with 5 views (each of 100 dimensions), each representing a molecular modality and labels (see Appendix D.1 for details). The binary label represents 1-year mortality of a patient-sample. We selected 2 views (mRNA and miRNA) which had 9,874 samples out of all possible combinations after filtering out samples with missing values. After adding data with missing labels to the training set, a 90%/5%/5% split was performed with 5 different seeds. Due to the complex nature of biological data, private information may be predictive as well.

As shown in Table 5, our method in general performs better than baseline methods in terms of accuracy (see Appendix D.3 for AUROC results), both learned shared and private latent spaces are predictive, and combining $\mathbf{z}$ and $\mathbf{w}$ achieves the best performance. This is likely because clean disentanglement separates predictive information between shared and private latent variables, making predictions based on combined latent space more robust. In particular, $\mathcal{L}_{\text{CrossMI}}$ contributed most to the performance, and adding diffusion priors in latent space consistently improves performance.

## 5 CONCLUSIONS

We have proposed IDMVAE, a generative model for learning disentangled representations from multimodal data. Our innovations include the incorporation of cross-view mutual information maximization for shared variable extraction, redundancy removal based on generative augmentation, and flexible latent priors with diffusion models. These components are complimentary to each other and jointly overcome the limitations of pure likelihood modeling, resulting in superior performance than existing state-of-the-art multimodal VAEs as well as non-generative disentanglement methods. In the future, we would like to extend the model to handle missing modalities (Lee & van der Schaar, 2021), leveraging the controllable generation capability of our model, and to scenarios where information is shared locally instead of globally among the views (Liu et al., 2023).

---

[3]https://www.cancer.gov/tcga

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

# A    DIFFERENT IMPLEMENTATIONS FOR GENERATIVE AUGMENTATION

In section 2.3, we have discussed two implementations of generative augmentation for redundancy removal. Here we provide a detailed comparison of them.

Recall that $\mathbf{x}_m$ and $\mathbf{x}'_m$ are two input samples, $(\mathbf{z}_m, \mathbf{w}_m)$ is pair of samples drawn from the posteriors $q(\mathbf{z}|\mathbf{x}_m)$ and $q(\mathbf{w}_m|\mathbf{x}_m)$ respectively, and similarly $(\mathbf{z}'_m, \mathbf{w}'_m)$ is a pair of samples drawn from conditional posteriors for $\mathbf{x}'_m$. With disentanglement and a good generative model, we could independently vary one variable while keeping the other the same to obtain a new sample. In particular, a sample $\mathbf{x}^+_m \sim p(\mathbf{x}_m|\mathbf{z}_m, \mathbf{w}'_m)$ would share the same $\mathbf{z}$ with $\mathbf{x}_m$. In turn, when we map $\mathbf{x}^+_m$ back to the latent space, $q(\mathbf{z}|\mathbf{x}^+_m)$ and $q(\mathbf{z}|\mathbf{x}_m)$ should be similar. Likewise, $q(\mathbf{w}_m|\mathbf{x}^+_m)$ and $q(\mathbf{w}_m|\mathbf{x}'_m)$ should be similar.

**Least squares matching.**    In the first implementation, we would like to minimize $I(\mathbf{z}_m; \mathbf{x}_n)$ by approximately minimizing $H(\mathbf{w}_m|\mathbf{x}'_m)$:

$$\mathbb{E}_{\mathbf{x}_m \sim p(\mathbf{x}_m), \mathbf{x}'_m \sim p(\mathbf{x}_m), \mathbf{z}_m \sim q(\mathbf{z}|\mathbf{x}_m), \mathbf{w}'_m \sim p(\mathbf{w}_m|\mathbf{x}'_m), \mathbf{x}^+_m \sim p(\mathbf{x}_m|\mathbf{z}_m, \mathbf{w}'_m)} [-\log q(\mathbf{w}'_m|\mathbf{x}^+_m)].$$

Assuming that posteriors are parameterized Gaussians, $\mathcal{L}_{\text{GenAug}, \mathbf{w}_m}$ reduces to $\ell_2$ loss for matching means of posteriors, and we implement it as

$$\mathcal{L}^{lsq}_{\text{GenAug}} = \mathbb{E}_{\mathbf{x}_m \sim p(\mathbf{x}_m), \mathbf{x}'_m \sim p(\mathbf{x}_m), \mathbf{z}_m \sim q(\mathbf{z}|\mathbf{x}_m), \mathbf{w}'_m \sim q(\mathbf{w}_m|\mathbf{x}'_m), \mathbf{x}^+_m \sim p(\mathbf{x}_m|\mathbf{z}_m, \mathbf{w}'_m)} \|\overline{\mathbf{w}}'_m - \overline{\mathbf{w}}''_m\|^2$$

where $\overline{\mathbf{w}}'_m$ is the posterior mean of $q(\mathbf{w}_m|\mathbf{x}'_m)$ while $\overline{\mathbf{w}}''_m$ is the posterior mean of $q(\mathbf{w}_m|\mathbf{x}^+_m)$.

**Contrastive matching.**    In practice, we find it more stable to use a contrastive loss for matching, i.e.,

$$\mathcal{L}^{contrast}_{\text{GenAug}} := -Contrast(\mathbf{w}''_m, \mathbf{w}'_m) \qquad \text{where} \quad \mathbf{x}^+_m \sim p(\mathbf{x}_m|\mathbf{z}_m, \mathbf{w}'_m), \ \mathbf{w}''_m \sim q(\mathbf{w}_m|\mathbf{x}^+_m).$$

We plug in the two different implementations into our loss. In Table 6 and 7, we provide the comparison of the two on PolyMNIST-Quadrant, each with its loss coefficient tuned on the validation set. We find the best coefficients to be $\lambda_1$=80 and $\lambda^{lsq}_2$=0.75 for $\mathcal{L}^{lsq}_{\text{GenAug}}$, and $\lambda_1$=80 and $\lambda^{contrast}_2$=20 for $\mathcal{L}^{contrast}_{\text{GenAug}}$; diffusion prior loss has a coefficient of 1.0 when incorporated. We observe that both implementations improve the disentanglement compared with using $\mathcal{L}_{\text{CrossMI}}$ only, with $\mathcal{L}^{contrast}_{\text{GenAug}}$ outperforming $\mathcal{L}^{lsq}_{\text{GenAug}}$.

Table 6: Comparison of $\mathcal{L}^{lsq}_{\text{GenAug}}$ and $\mathcal{L}^{contrast}_{\text{GenAug}}$ for generative augmentation regularization in latent linear classification on PolyMNIST-Quadrant. Accuracies are averaged over 5 modalities.

| Our Models | $z \rightarrow Digit \uparrow$ | $z \rightarrow Quad \downarrow$ | $w \rightarrow Quad \uparrow$ | $w \rightarrow Digit \downarrow$ |
|---|---|---|---|---|
| $\mathcal{L}_{\text{CrossMI}}$ Only ($\lambda_2 = 0$) | 0.977 | 0.277 | 0.999 | 0.202 |
| $\mathcal{L}_{\text{CrossMI}} + \mathcal{L}^{lsq}_{\text{GenAug}}$ | 0.972 | 0.267 | 0.999 | 0.186 |
| + diffusion prior | 0.980 | **0.263** | 0.999 | 0.154 |
| $\mathcal{L}_{\text{CrossMI}} + \mathcal{L}^{contrast}_{\text{GenAug}}$ | **0.983** | 0.271 | 0.999 | 0.162 |
| + diffusion prior | 0.982 | 0.267 | 0.999 | **0.143** |

Table 7: Comparison of $\mathcal{L}_{\text{GenAug}}^{lsq}$ and $\mathcal{L}_{\text{GenAug}}^{contrast}$ for generative augmentation regularization in generative coherence, averaged over 5 views, on PolyMNIST-Quadrant. We use subscript $q$ to indicate samples from posteriors and subscript $p$ to indicate samples from priors. For generated images, digit label is determined by $\mathbf{z}_{s,q}$ or otherwise random (with target accuracy 10%), quadrant label is determined by $\mathbf{w}_{s,q}$ or otherwise random (with target accuracy 25%).

| Our Models | Self Gen ($s = t$) | | | | Cross Gen ($s \neq t$) | Uncond. |
|---|---|---|---|---|---|---|
| | $Gen(\mathbf{z}_{s,q}, \mathbf{w}_{t,p})$ | | $Gen(\mathbf{z}_{t,p}, \mathbf{w}_{s,q})$ | | $Gen(\mathbf{z}_{s,q}, \mathbf{w}_{t,p})$ | $Gen(\mathbf{z}_p, \mathbf{w}_p)$ |
| | $Digit \uparrow$ | $Quad \downarrow$ | $Digit \downarrow$ | $Quad \uparrow$ | $Digit \uparrow$ | $Digit \uparrow$ |
| $\mathcal{L}_{\text{CrossMI}}$ Only ($\lambda_2 = 0$) | 0.670 | 0.250 | 0.370 | 0.999 | 0.671 | 0.008 |
| $\mathcal{L}_{\text{CrossMI}} + \mathcal{L}_{\text{GenAug}}^{lsq}$ | 0.817 | 0.250 | 0.219 | 0.999 | 0.812 | 0.044 |
| + Diffusion prior | 0.917 | 0.249 | 0.109 | 0.999 | 0.875 | **0.668** |
| $\mathcal{L}_{\text{CrossMI}} + \mathcal{L}_{\text{GenAug}}^{contrast}$ | 0.898 | 0.249 | 0.162 | 0.999 | 0.881 | 0.070 |
| + Diffusion prior | **0.942** | 0.251 | **0.106** | 0.999 | **0.887** | 0.664 |

# B    DETAILS AND ADDITIONAL RESULTS ON POLYMNIST-QUADRANT

## B.1    IMPLEMENTATION DETAILS

We utilize a deep residual network (ResNet) architecture of 3 residual blocks, with the number of filters doubling from 64 to up to 512 after each block for the encoder, for all five modalities. Our model has a total of 201M parameters without diffusion prior and 206.5M Parameters with diffusion prior. And each modality's information is factorized in the latent space into a shared latent dimension of 32 and a private latent dimension of 128. Models are trained for 100 epochs using the Adam optimizer with a learning rate of $5e^{-4}$ and a batch size of 128, and use the other default hyperparameters of MMVAE+ baseline, including the KL divergence coefficient $\beta$ of 2.5. We performed a grid search over the coefficients to tune the regularization terms, $\lambda_1$ and $\lambda_2$, after training for 100 epochs. We search them in the range $[0.01, 100]$. We tune $\lambda_1$ individually first to find the best general performance in latent classification for $\mathcal{L}_{\text{CrossMI}}$, and fix the $\lambda_1$, then combine with $\mathcal{L}_{\text{GenAug}}^{contrast}$, and find the best combination of $\lambda_1$=80 and $\lambda_2^{contrast}$=20. Finally, we tune the diffusion prior weight to 1.0 out of $\{0.01, 0.1, 1.0, 10.0\}$, which optimizes the final general performance at the 100th epoch.

## B.2    CONDITIONAL GENERATION

In Figure 6 and Figure 7, we provide additional results on conditional generations, where one latent variable is sampled from the posterior, while the other is sampled from the prior. In Figure 8, we provide conditional generations for which both $\mathbf{z}$ and $\mathbf{w}$ are sampled from posteriors; this simulates the samples we use in $\mathcal{L}_{\text{GenAug}}$. In Table 8, we give a quantitative measure of the generation results using FID (Heusel et al., 2017). In all cases, our method provides the most coherent generations, consistent with the quantitative results in Section 4.1.

Table 8: Generative quality, as measured by FID, on PolyMNIST-Quadrant test set.

| Model | Cross Conditional FID $\downarrow$ | Uncondidtional FID $\downarrow$ |
|---|---|---|
| DMVAE | 100.817 | 79.646 |
| MMVAE+ | 86.091 | 87.008 |
| SBM-VAE | 128.7 | 128.8 |
| IDMVAE (ours) | 84.528 | 87.108 |
| $-\mathcal{L}_{\text{CrossMI}}$ ($\lambda_1 = 0$) | 85.589 | 85.563 |
| $-\mathcal{L}_{\text{GenAug}}$ ($\lambda_2 = 0$) | 84.698 | 85.596 |
| + Diffusion prior | **73.186** | **73.681** |

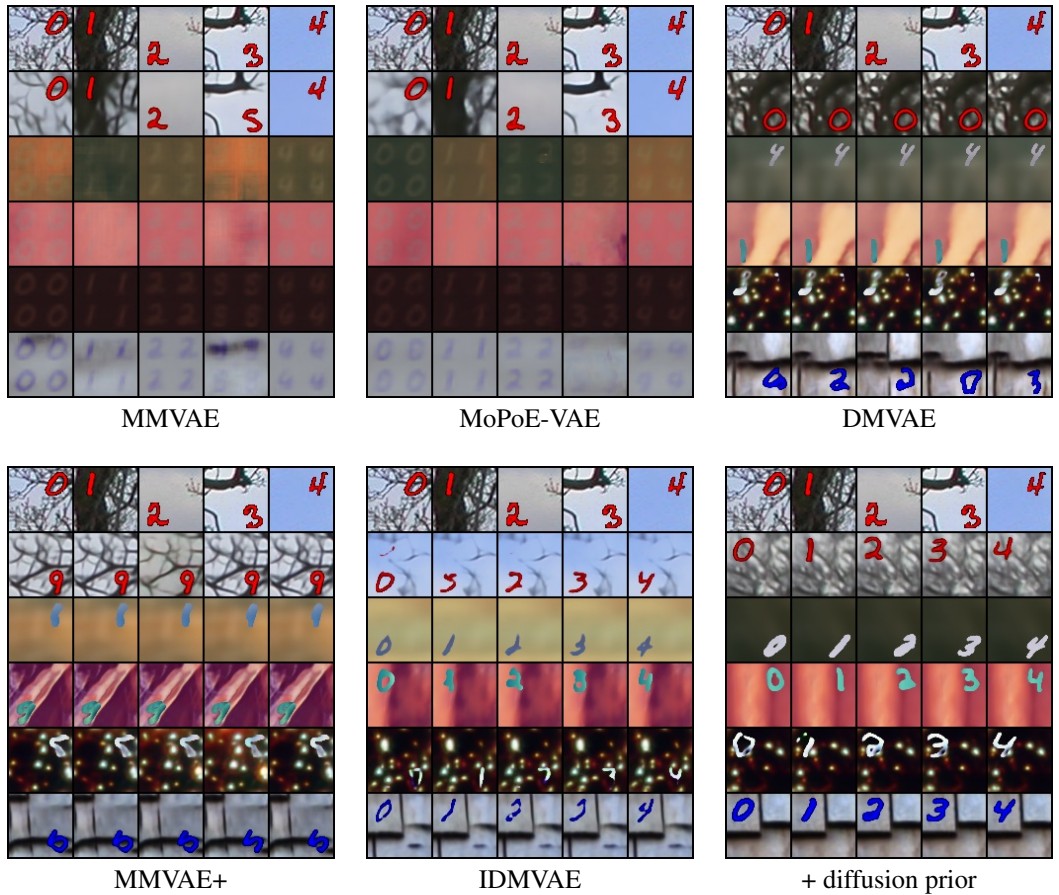

Figure 6: Conditional generations on PolyMNIST-Quadrant, with conditioning on $\mathbf{z}$. The top row shows the samples (from modality 1) we condition on. We sample $\mathbf{z} \sim q(\mathbf{z}|\mathbf{x}_1)$, sample the private variable from the corresponding prior $\mathbf{w}_m \sim p(\mathbf{w}_m|\mathbf{x}_m)$, and generate a new sample from $p(\mathbf{x}_m|\mathbf{z}, \mathbf{w}_m)$. Row 2 to row 6 are generated samples for modalities 1 to 5. Note for well-disentangled latent variables, each column shall contain the same digit $\mathbf{z}$. For each row, we used the same prior sample of $\mathbf{w}$, so images in the same row shall have the same quadrant, writing style, and background.

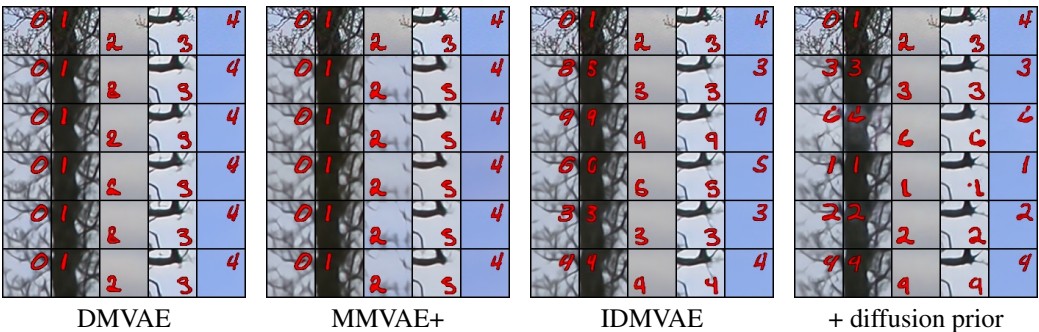

Figure 7: Conditional generations on PolyMNIST-Quadrant, with conditioning on $\mathbf{w}_m$. The top row shows the samples (from modality 1) we condition on. We sample $\mathbf{w} \sim q(\mathbf{w}|\mathbf{x}_1)$, and sample $\mathbf{z} \sim p(\mathbf{z})$, and generate a new sample from $p(\mathbf{x}_1|\mathbf{z}, \mathbf{w}_1)$. Row 2 to row 6 are generated samples. Note for well-disentangled latent variables, each column shall have the same quadrant position and background. For each row we used the same prior sample of $\mathbf{z}$, so images in the same row shall have the same digit.

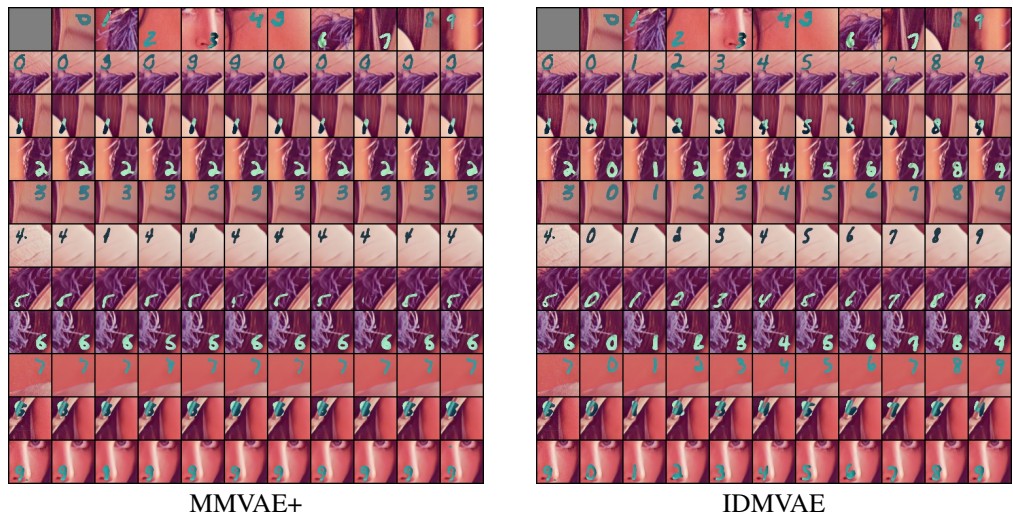

MMVAE+             IDMVAE

Figure 8: Conditional generation on PolyMNIST-Quadrant. The first row and first column contain images for which we extract posterior samples of $\mathbf{z}$ and $\mathbf{w}_m$, respectively. And the rest of the grid contains generated images using latent samples of the corresponding row and column. This figure illustrates the samples we use in generative augmentation.

## B.3 Latent Visualization

In Figure 9 and Figure 10, we provide 2D visualizations of latent representations of view 2 (i.e., samples of $q(\mathbf{z}|\mathbf{x}_2)$ and $q(\mathbf{w}|\mathbf{x}_2)$, respectively) on the PolyMNIST-Quadrant test set. Our method leads to improved separation of digit or quadrant classes in the latent spaces, as also shown in the quantitative analysis (Table 1). Observe that, without a diffusion prior, there exists a gap between the posterior and the Gaussian prior, whereas the capacity of the diffusion prior is strong enough to ensure good overlap between the two distributions, and therefore leads to superior unconditional generation.

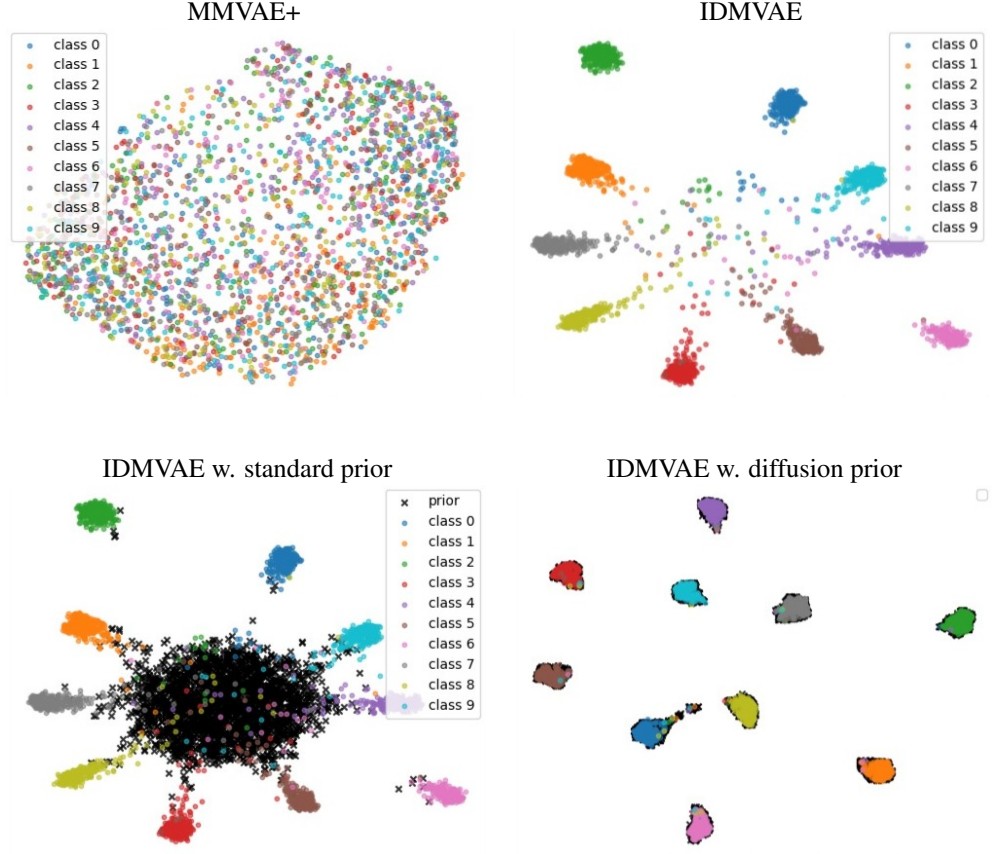

Figure 9: 2D visualization (by UMAP) of learned $\mathbf{z}_2$ on the test set. We color each point according to its ground truth digit label, and black markers correspond to samples from the prior.

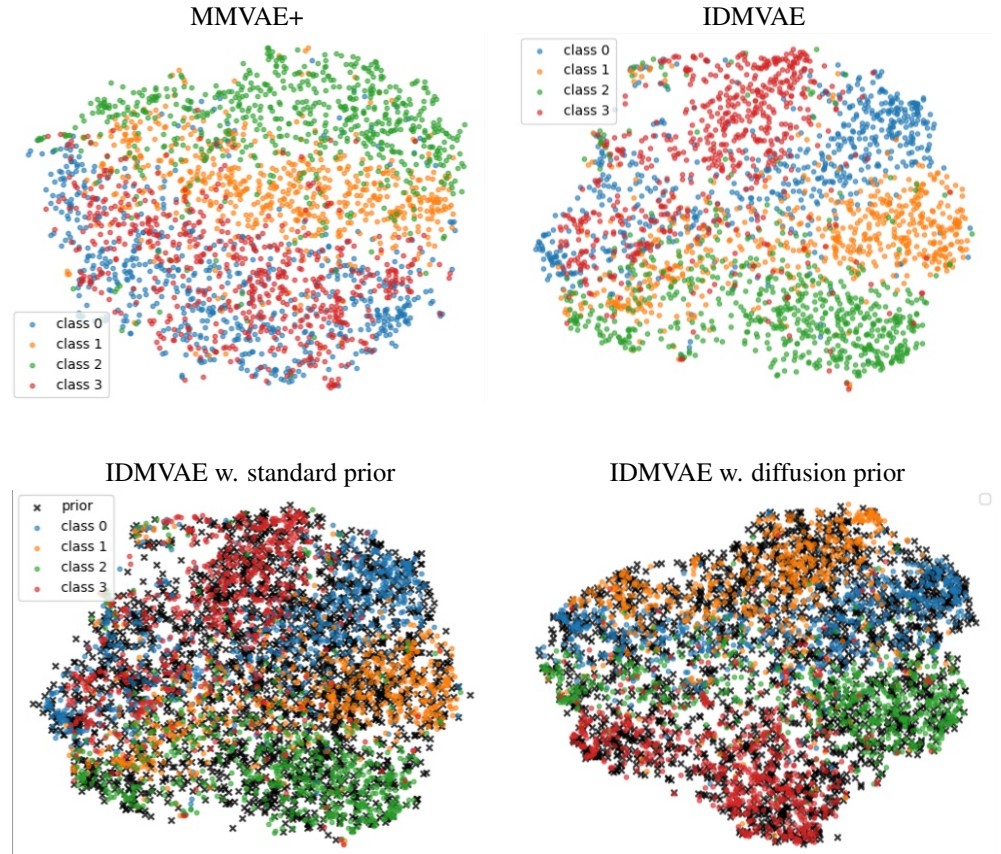

Figure 10: 2D visualization (by UMAP) of learned $\mathbf{w}_2$ on the test set. We color each point according to its ground truth quadrant label, and black markers correspond to samples from the prior.

## C    RESULTS ON HIGH-QUALITY CUB (CUB-HQ)

### C.1    DATASET

The 8 super-categories, namely Blackbird, Gull, Jay, Oriole, Tanager, Tern, Warbler, and Wren, are created following the same grouping method introduced by Palumbo et al. (2024), as shown in Figure 11. In addition, we introduce a private binary label representing the bird's horizontal direction, determined by the part location annotations provided in the original dataset (Wah et al., 2011), as illustrated in Figure 12. Specifically, we compare the average horizontal position of the group of the bird's 'head' parts with the average horizontal position of the group of its 'body' parts. If the head is positioned to the left of the body, the direction label is 'left' (label 0); otherwise, it is 'right' (label 1). This creates a modality-specific (private) label for the image that cannot be inferred from the text captions. At the same time, as shown in Figure 12 (b) and (c), a very small fraction of the images have invisible 'head' or 'body' location annotations, or the locations are too close, in which case they are not assigned the direction label. Direction labels of validation and test images are verified by human.

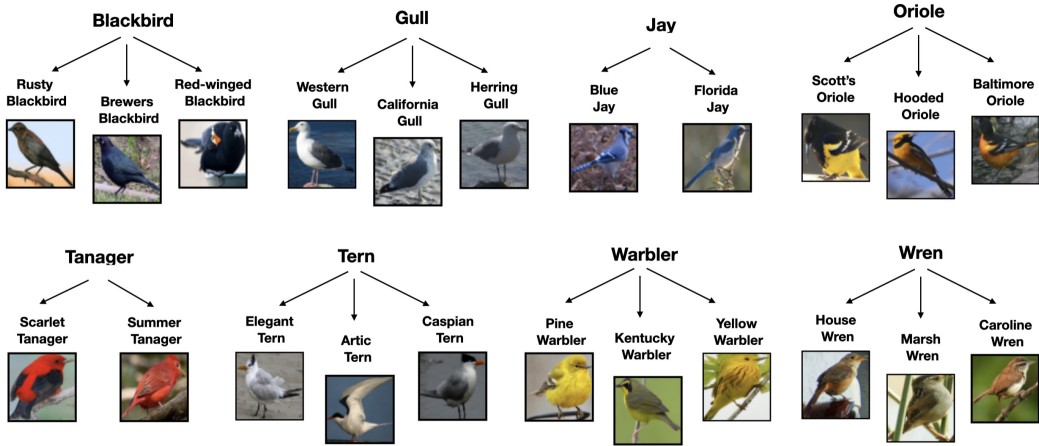

Figure 11: CUB category labels: dividing 22 species into 8 super-categories (Palumbo et al., 2024).

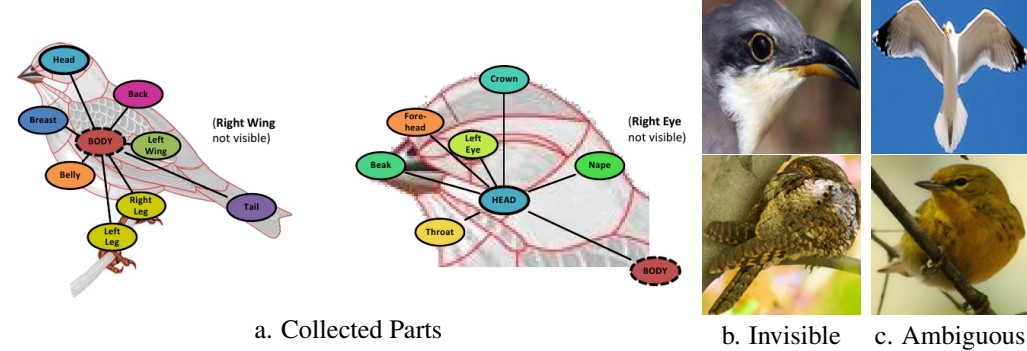

a. Collected Parts

b. Invisible    c. Ambiguous

Figure 12: The collected parts in the original dataset (Wah et al., 2011), and sample images with invisible 'body' or 'head' location annotations or ambiguous horizontal direction.

In this experiment, we use the 256x256 resolution version of the original CUB dataset (Wah et al., 2011) *without* cropping images using bounding boxes for birds. Compared to the CUB dataset used in the main paper, the CUB-HQ setup contains richer background and is more challenging for representation learning.

## C.2 Implementation Details

**Pre-trained VAE for data pre-processing**   We pre-process the image modality with a pretrained-VAE encoder[4] similarly used by Peebles & Xie (2022), converting each original RGB images to a $4 \times 32 \times 32$ tensor, which then serves as the model input. We can then apply the corresponding pre-trained decoder on samples of our model to generate high resolution images.

**Architecture of IDMVAE**   We use a deep residual network (ResNet) for the image modality. Both the encoder and decoder have five residual blocks. For the encoder, the number of filters doubles after each block, starting at 64 and ending at 1024. The decoder mirrors this, with the number of filters halving after each block. For the text modality, we utilize a CNN-based encoder and decoder on one-hot encoded captions with a vocabulary size of 1,590 words. Our model has a total of 137M parameters without diffusion prior and 140M Parameters with diffusion prior. During training, we apply horizontal flip augmentation to the image modality, with a flip probability of $0.5$.

**Hyperparameters**   Both the shared variable and the private variable have 256 dimensions. Models are trained for 50 epochs using Adam optimizer with a learning rate of $10^{-4}$ and a batch size of 256, with the rest of the hyperparameters being the default of the MMVAE+ baseline, including the KL divergence coefficient $\beta = 1.0$.

Then we tune the coefficients of $\mathcal{L}_{\text{CrossMI}}$, $\mathcal{L}_{\text{GenAug}}$, and diffusion prior similarly to B.1, and obtain the optimal coefficients $\lambda_1 = 40$, $\lambda_2 = 10$, and diffusion loss weight $0.1$.

**High resolution image generation**   After training, our model can generate images of 4 channels and $32 \times 32$ resolution, which can then be decoded using the pretrained-VAE decoder into $256 \times 256$ RGB images. However, given the relatively small training set size of CUB, the generated images tend to be blurry. Following Pandey et al. (2022) and Palumbo et al. (2024), we train a diffusion model to generate high-quality images conditioned on our model's generations. Specifically, for each model, we extract its reconstructions of the training data by the image modality architecture (i.e., applying two encoders to obtain $(\mathbf{z}_0, \mathbf{w}_0)$, followed by the image decoder), and train a DiT model (Peebles & Xie, 2022) to generate the ground-truth training images from pure Gaussian noise, conditioned on model reconstructions. We use the pre-trained checkpoint of the XL-2 architecture, and add an additional patch embedding layer to map conditioning features into representations that are later incorporated into the AdaLN module of each DiT block. We finetune all parameters of the resulting model, with minibatches of 32 images for 70000 updates.

## C.3 Results

**Generative coherence**   To measure the generative coherence of models, we perform image-to-text, text-to-image, and image-to-image generations. We measure the quality of generated high-resolution images with FID (Heusel et al., 2017), and the coherence between text and images using CLIPScore (Hessel et al., 2021). These metrics are given in Table 4. For reference, the CLIPScore between ground truth test images and text is 0.762. The generation performance of our model is superior to those of the baselines. We provide samples of text-to-image generation in Figure 13, samples of image-to-text generation in Figure 14, and samples of image-to-image generation in Figure 15 and Figure 16.

**Generative augmentation**   To understand how generative augmentation and $\mathcal{L}_{\text{GenAug}}$ help with disentanglement, we visualize such generations in Figure 17. The first row and first column contain images for which we extract posterior samples of $\mathbf{z}$ and $\mathbf{w}$, respectively. And the rest of the grid contain generated images using samples of $\mathbf{w}$ of the corresponding row and $\mathbf{z}$ of the corresponding column. We observe that images in each column mostly share the same bird color, while images in each row mostly share the same orientation as desired. It is important to note that, different from PolyMNIST, the generated images with our ResNet architecture are not of high resolution, but they already capture essential information regarding shared and private labels. Conditional generations by DiTs are of high quality, showing that we can independently vary shared and private variables to obtain controllable generations for complex modalities.

---

[4]https://huggingface.co/stabilityai/sd-vae-ft-mse

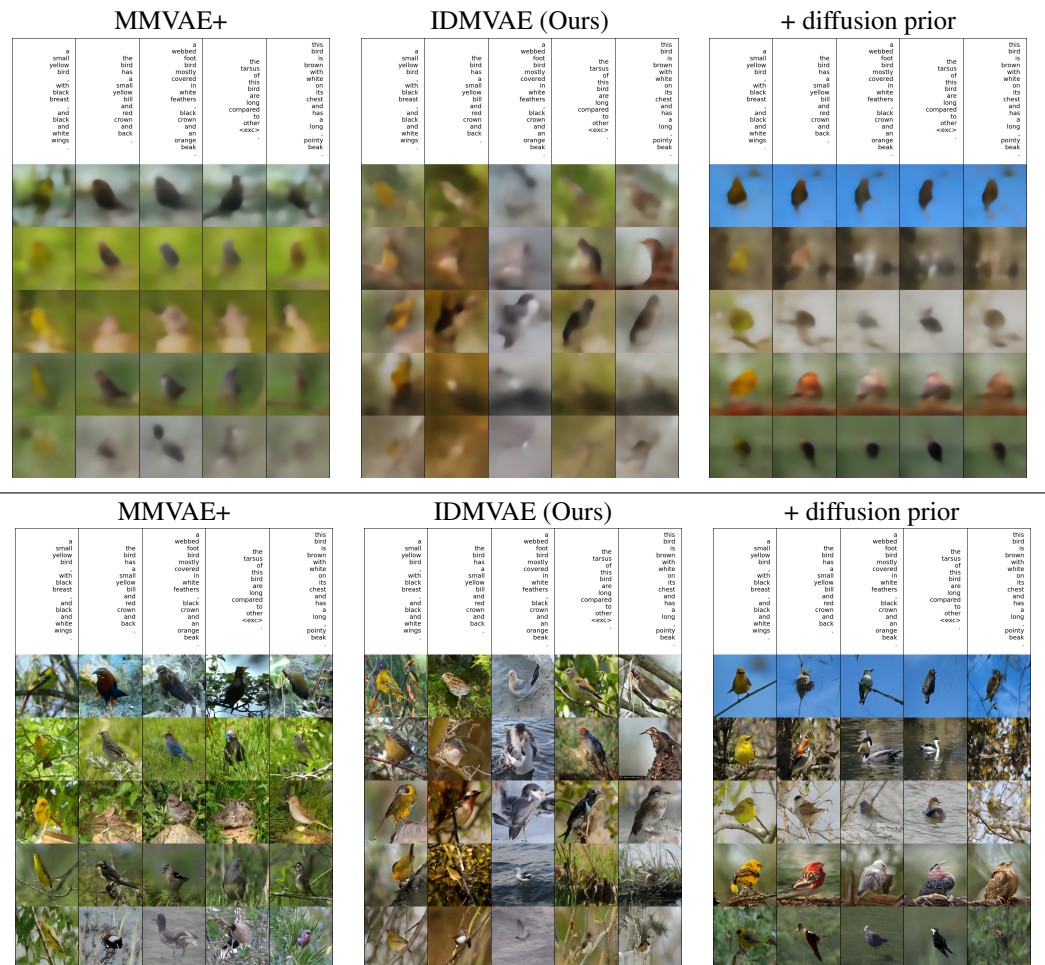

Figure 13: Text-to-image generation on CUB-HQ, without DiT conditional generation (top panel) and with DiT conditional generation (bottom panel). We combine the posterior sample **z** of the text modality (top row) with the prior sample **w** of the image modality (shared by each row) for generation. Ideally, images in the same column are of similar bird category and color, and images in the same row shall have the same bird orientation.

Figure 14: Image-to-text generation on CUB-HQ. We combine the posterior sample **z** of the image modality (top row) with the prior sample **w** of the text modality (shared by each row) for generation.

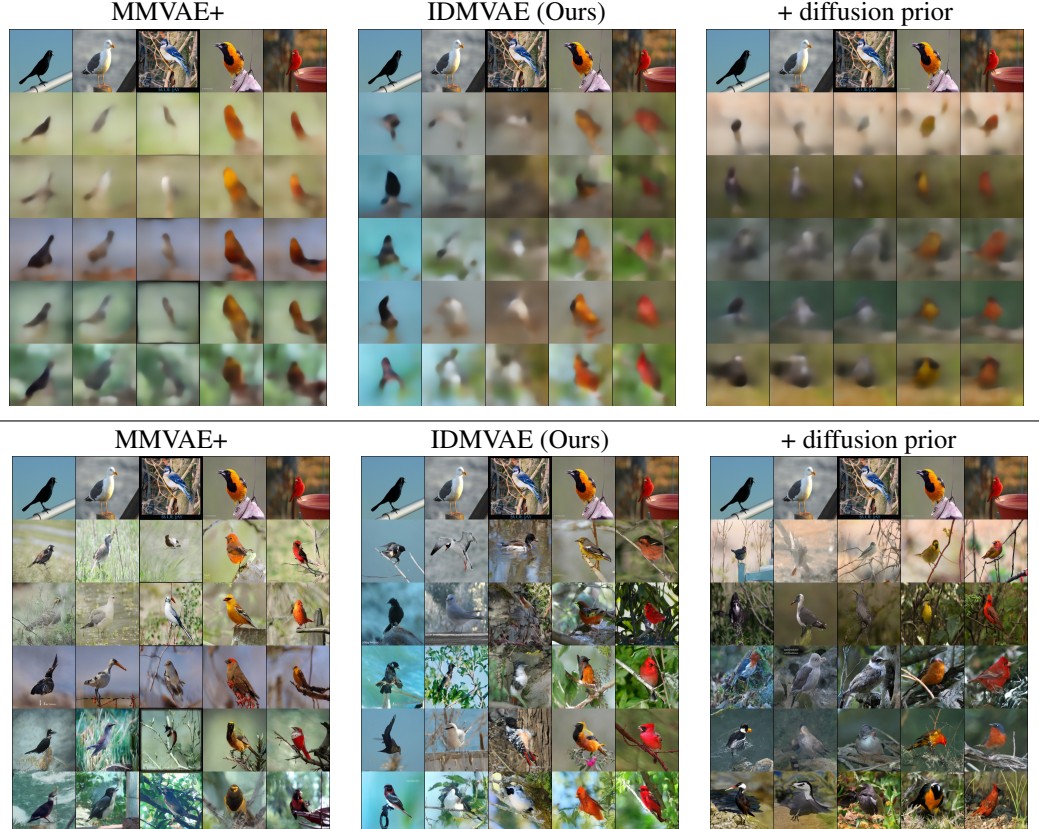

Figure 15: Image-to-image generation on CUB-HQ, without DiT conditional generation (top panel) and with DiT conditional generation (bottom panel). We combine the posterior sample $\mathbf{z}$ of the image modality (top row) with the prior sample $\mathbf{w}$ of the image modality (shared by each row) for generation. Ideally, images in the same column are of similar bird category and color, while images in the same row shall have the same bird orientation.

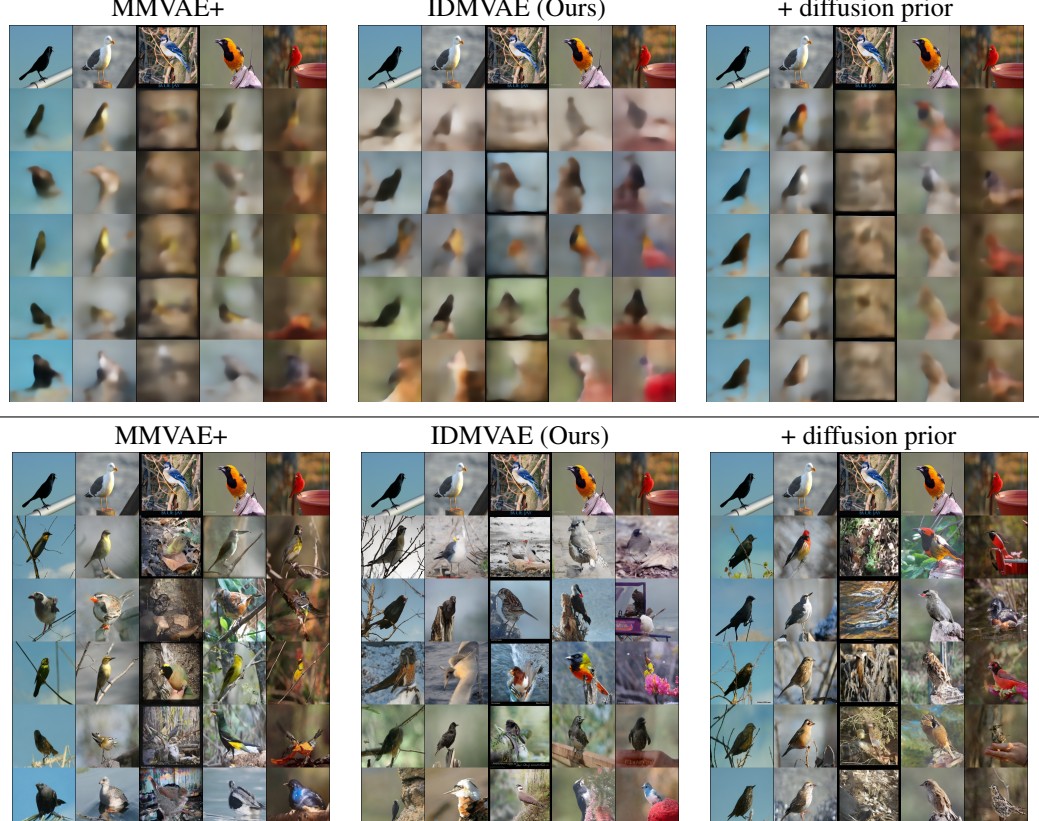

Figure 16: Image-to-image generation on CUB-HQ, without DiT conditional generation (top panel) and with DiT conditional generation (bottom panel). We combine the posterior sample **w** of the image modality (top row), with the prior sample **z** of the image modality (shared by each row) for generation. Ideally images in the same column are of the same bird orientation, while images in the same row shall have similar bird category and color.

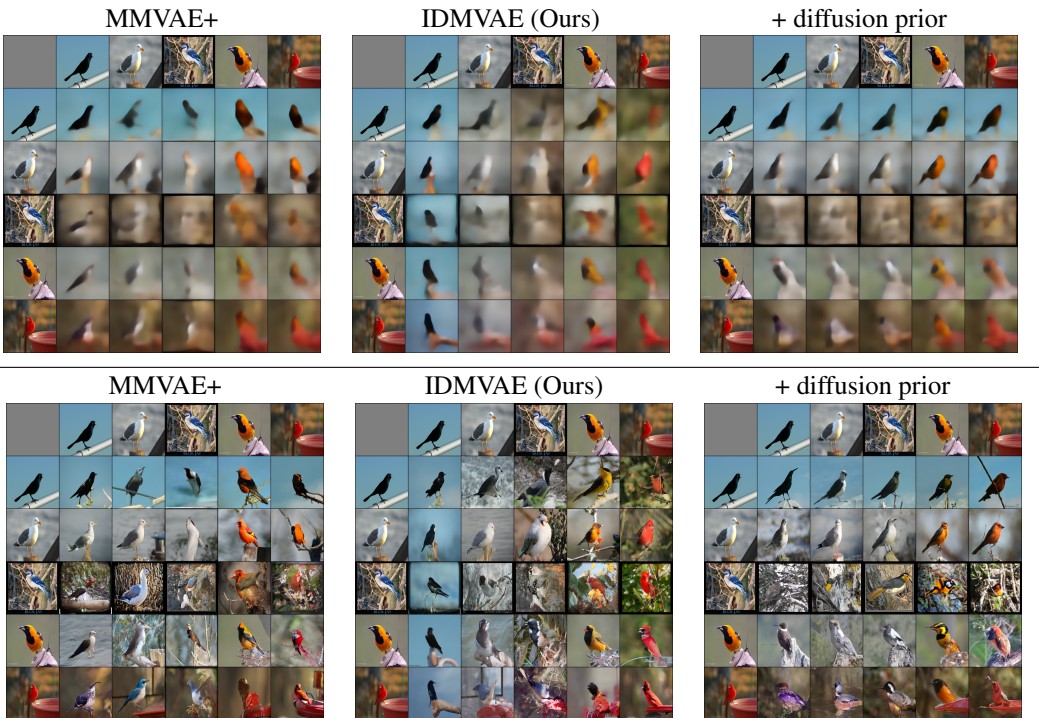

Figure 17: Generative augmentations on CUB-HQ, without DiT conditional generation (top panel) and with DiT conditional generation (bottom panel). The first row and first column contain images for which we extract posterior samples of **z** and **w**, respectively. And the rest of the grid contain generated images using samples of **w** of the corresponding row and **z** of the corresponding column. Ideally, images in the same column are of the same bird category and color, while images in the same row shall have the same bird orientation.

**Latent Visualization**    In Figure 18 and Figure 19, we provide a visualization of the latent representations of shared variables. Consistent with latent classification results (Table 3), our method leads to better separation of classes. Furthermore, diffusion priors are flexible enough to model the structured latents.

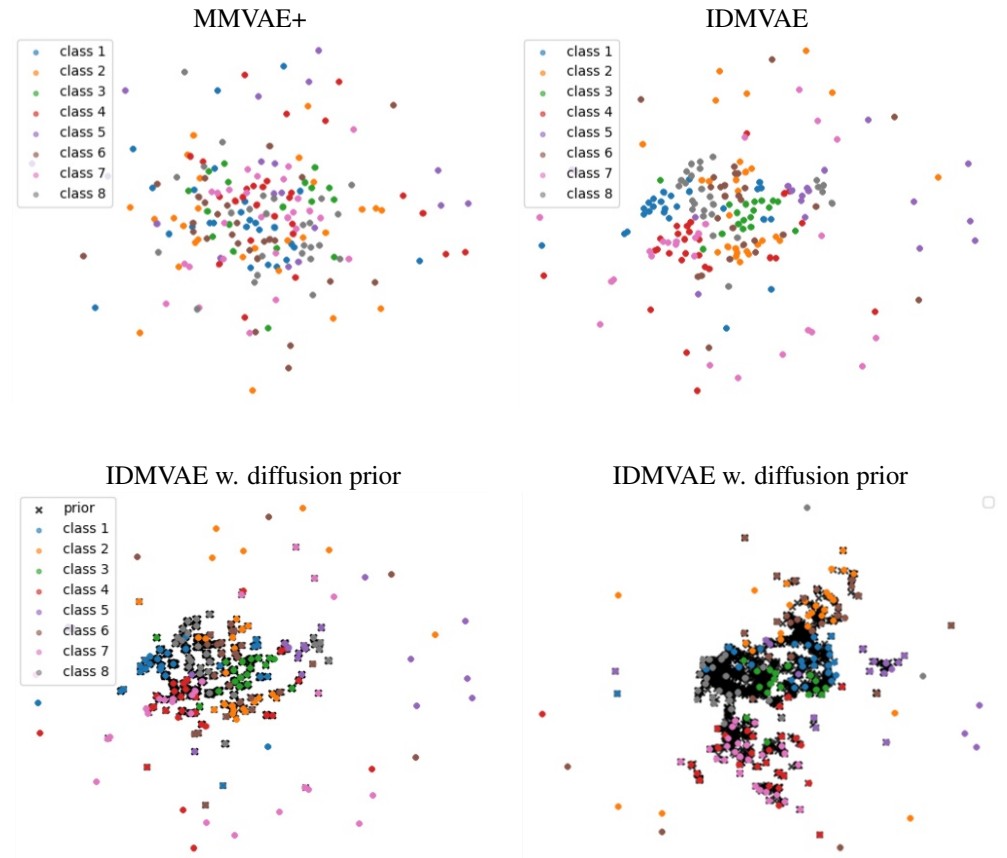

Figure 18: 2D visualization (by UMAP) of $z_1$, i.e., learned shared representation of the image modality, on the validation set. We color each representation according to its ground truth bird category cluster label, and black markers correspond to samples from the prior.

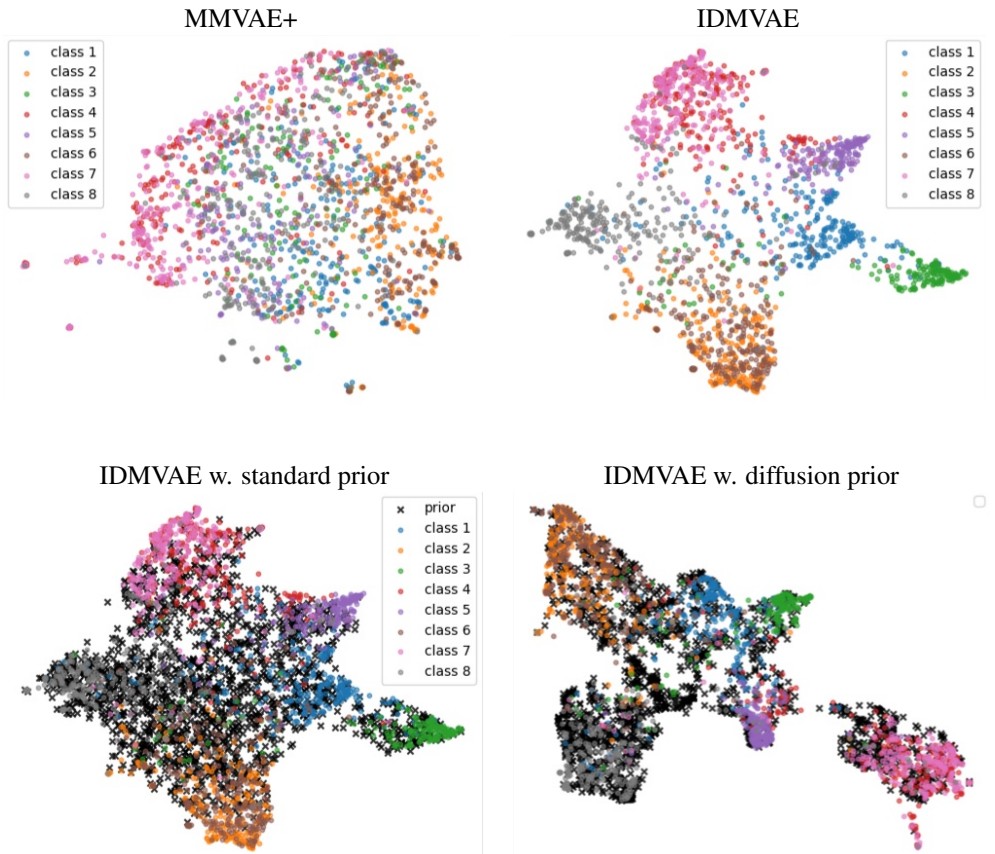

Figure 19: 2D visualization (by UMAP) of $\mathbf{z}_2$, i.e., learned shared representation of the text modality, on the validation set. We color each representation according to its ground truth bird category cluster label, and the black markers correspond to samples from the prior.

# D  DETAILS AND ADDITIONAL RESULTS ON TCGA

## D.1  DATA PREPARATION

TCGA dataset (2016 version) is processed by combining multi-omics data across different cancer types and different patients before features are selected and kernel PCA was performed to reduce the dimensionality to 100. Because clinical trials usually have small sample sizes, high dimensionalities, and complex dependencies, this dataset is ideal to test the robustness of our method. One significant outcome of cancer multi-omics data is the days of survival after samples were collected. Because patients may not show up for checkups, it is possible that a sample is censored (unlabeled). The days of survival are then converted to a binary 1-year mortality indicator. The dataset itself has missing views but contains predictive information and high correlations among different views, making it suitable for our task. Further notice that this dataset does not contain private ground truth.

## D.2  IMPLEMENTATION

A 2-layer MLP with 128 as hidden dimensions was used for encoding and decoding with 48 latent dimensions (16 for $\mathbf{z}$ and 32 for $\mathbf{w}$). Evaluation for all methods regarding this dataset is done by averaging logits from each view. 50 epochs were run to train the model. For TCGA dataset, baseline methods are performed with default hyperparameter, which gives KL divergence a coefficient of 2.5. For DisentangledSSL baseline in particular, step 1 coefficient was set to 0 since we use posterior mean instead of zsample to match other methods and step 2 coefficient was set to 0.01. To tune our method, we performed a grid search with coefficients $\{0.001, 0.01, 0.1, 1, 10, 100\}$ and chose the best combination on validation set, before recording the performance on test set. $\lambda_1 = 10, \lambda_2 = 0.001$ were chosen to be the best combination at 40 epochs. For ablation studies, we set one coefficient to be 0 while keeping the other one optimal in a combined setting. For the optimal coefficients combination, we used the model at epoch 40; $\lambda_1 = 0$, at epoch 50; and $\lambda_2 = 0$, at epoch 35. For adding a diffusion prior, we tuned the diffusion weight to be 0.1 out of 0.1, 1, 10 while keeping $\lambda_1, \lambda_2$ same as the optimal combination and chose the best performance at validation set at epoch 40.

## D.3  PREDICTION AUROC

In Table 9, we provide the linear classification AUROC of different methods using latent representations. The relative merits of different methods are consistent with that observed with the accuracy metric in Table 5.

Table 9: Prediction AUROC Performance with ablation on TCGA dataset, averaged over 2 modalities and 5 splits. Tuning reported in Appendix D.2.

| Model | $\mathbf{z}$ ↑ | $\mathbf{w}$ | $\mathbf{z} + \mathbf{w}$ ↑ |
|---|---|---|---|
| MMVAE | 0.653±0.033 | — | — |
| MoPoE-VAE | 0.660±0.024 | — | — |
| DMVAE | 0.609±0.030 | 0.636±0.037 | 0.643±0.032 |
| MMVAE+ | 0.586± 0.027 | 0.581±0.033 | 0.585±0.033 |
| DisentangledSSL | 0.693±0.046 | 0.551±0.019 | 0.699±0.045 |
| IDMVAE (ours) | 0.740±0.025 | 0.740±0.022 | 0.767±0.026 |
| − $\mathcal{L}_{\text{CrossMI}}$ ($\lambda_1 = 0$) | 0.549±0.017 | 0.545±0.026 | 0.548±0.026 |
| − $\mathcal{L}_{\text{GenAug}}$ ($\lambda_2 = 0$) | 0.740±0.019 | 0.746±0.022 | 0.771±0.021 |
| + Diffusion prior | **0.745±0.024** | **0.751±0.029** | **0.772±0.022** |

