# OpenReview forum: "Disentanglement of Variations with Multimodal Generative Modeling"
_ICLR.cc/2026/Conference — ICLR 2026 Poster_

### Official Review · Reviewer_wgHU · 2025-10-25

**Soundness:** 3
**Presentation:** 2
**Contribution:** 3
**Rating:** 4
**Confidence:** 4

**Summary:**

This paper introduces Information-Disentangled Multimodal VAE (IDMVAE), a new multimodal generative framework designed to improve disentanglement between shared and modality-specific representations.

**Strengths:**

1. The cross-view regularization for disentangling the shared latent variable z and modality-specific w is well-motivated.

2. The experiments consider cross-domain data, and the ablation studies are complementary.

3. Using diffusion to model latent priors results in better alignment between prior and posterior distributions, leading to improved unconditional generation and coherence.

**Weaknesses:**

1. Compared to multimodal VAEs, the novelty is based on the cross-view regularization. For the diffusion part, CMVAE also integrates the diffusion model for improved performance. The conceptual advance is relatively modest.

2. The paper does not provide training efficiency comparisons or runtime analyses (especially for the diffusion part). It unclear whether the performance comes from the model complexity.

3. While figures illustrate disentanglement, generated samples on CUB are of relatively low fidelity, suggesting the model’s generative capacity is still limited.

**Questions:**

1. Can you provide a way to visualize the learned shared latent variable and modality-specific variables? Comparing the visualizations for the MMVAE+ and the proposed method could be straightforward.

2. Can this method scale up to a large-scale Multimodal dataset (e.g., MSCOCO)?

3. How does the proposed method compare to other diffusion models in terms of, for example, CLIP score?

4. How do you handle the text input? Any pre-trained embedding models?

---

> ### Author Response · Authors · 2025-11-26
>
> Dear reviewer,
>
> We have significantly revised the experiments, including a new dataset CUB-HQ which has high resolution and more background  and is therefore more challenging.
>
> 1. We have visualizations of latent representations in Figure 9, 10, 21, 22. Observe that our method achieves good separation of classes without hard supervision, and diffusion priors matches VAE posteriors.
>
> 2. With the help of DiT denoiser, our method achieves high quality generations for 256x256 resolution, see Figure 16, 18, 19, 20. We provide FID and CLIPScore when possible and our method outperforms important baselines.
>
> 3. We have not use pretrained embedding for text, we use one-hot representations as inputs and a CNN on top; this architecture is consistent with the MMVAE+ baseline. Using pretrained model will likely improve our performance.
>
> 4. Training efficiency:
>
> - We have now added model parameters in the revised paper. Our IDMVAE model for CUB-HQ has 140M parameters, and for PolyMNIST our model has 200M parameters.
>
> - We shall think of the complexity of our model as forwarding multiple VAEs (with mixture-of-expert aggregation scheme we perform reconstruction between all pairs of views); the CrossMI loss is efficient as it is simply computing contrastive loss in latent space without forwarding additional module beyond the VAE encoders; the GenAug loss is more costly as it generates a new set of inputs and then encode them; diffusion priors only work in latent space with a DNN. Overall the method costs less than double the cost of MMVAE+ (because GenAug is performed within each view and not across views).
>
> - Note for CUB-HQ, we have leveraged pretrained CNN ("stabilityai/sd-vae-ft-mse") for reducing the input resolution to 32x32, and finetuned pretrained DiT for denoising the IDMVAE samples (with small amount of steps and data). This shows that with clever use of pretrained models, the IDMVAE model is efficient and therefore scalable to larger dataset. We train our IDMVAE model on a single H100 and it took 17 hours for 50 epochs on the CUB-HQ dataset (110K image/caption pairs).
>
> - Our Figure 20 also shows that, it may not be necessary to use DiT to produce high-resolution in GenAug during representation learning, because the IDMVAE samples (top panel) already captures essential information (color of bird and direction of bird) and achieved disentanglement. A DiT that is finetuned separately can then fill in the details for high quality generation (bottom panel).
>
> - We think that as a paper that proposes a new algorithm, our datasets are realistic (which are also widely used in multi-modal VAE papers in top ML venues for the past two years) and experiments are convincing. It provides new insights for large-scale mutli-modal pretraining that can be pursued in the future.

---

### Official Review · Reviewer_bE3u · 2025-10-26

**Soundness:** 2
**Presentation:** 3
**Contribution:** 2
**Rating:** 2
**Confidence:** 5

**Summary:**

This paper introduces the Information-disentangled Multimodal VAE (IDMVAE), a new model for learning from data with multiple modalities. IDMVAE achieves this improved disentanglement by using Mutual Information (MI) regularizations and latent diffusion priors. They try to learn a common latent z that captures all the shared information encouraged by the added new regularization losses introduced in this work. In addition, they utilize diffusion models to create more complex and capable latent priors, thereby moving beyond simple Gaussian priors and enhancing the model's representational capacity. The authors demonstrate that IDMVAE achieves better shared and private information, resulting in superior generation quality and enhanced semantic coherence on some simple datasets compared to existing methods.

**Strengths:**

The paper is well-written and well-presented. The paper also seems to have strong theoretical grounds for the idea they presented in this work. Their MI-based regularization methods could be useful for multimodal VAE works in general. The proposed losses are also useful for learning a shared representation, which is important for downstream tasks. Their proposed idea of using practical contrastive loss for cross-view MI maximization and their disentanglement learning with generative augmentation losses is a good contribution. They propose using diffusion models for learning complex latents, but this has already been proposed and implemented in other works (look at the weaknesses part).

**Weaknesses:**

Even though the paper is well presented, there are considerable weaknesses in this paper that render the contribution marginal.


1. $ \textbf{Used datasets are toy datasets} $

The datasets used in this work are very small and not significant enough to show the practicality of the presented method. I encourage the authors to look at [1], which is also a work done on multimodal VAEs using a score-based model that used the CelebAMask-HQ multimodal dataset, which is larger than the datasets used in this paper, as a starting point. The paper is also not cited and not compared as a baseline. The authors can also add other larger datasets than this to prove their method is still useful in these settings.

2. $\textbf{Latent diffusion proposal}$

 Using a latent diffusion/score-based model to learn complex latents for multimodal VAEs is also presented in [2]. But it hasn't been cited in this work and presented as a new thing.

3. $\textbf{Limited baselines}$

 The baselines compared in this work are limited. Additional recent baselines should also be added to the work. E.g., [1,2,3,4].

4. $\textbf{Evaluation of Quality}$

 The paper doesn't use FID for evaluating the quality of generated unconditional and conditional images as in [1]. I believe a quantifiable measure of quality is important in multimodal VAE works.

5. $\textbf{Evaluation of method of PoE Multimodal Models}$

 The paper proposes that the method can be used in PoE/MoPoE multimodal VAE models but those haven't been explored as ablations. Can the authors try their method in at least one setup of these multimodal VAEs as a proof of concept or ablation study.


References
1. Wesego, Daniel, and Amirmohammad Rooshenas. "Score-based multimodal autoencoders." arXiv preprint arXiv:2305.15708 (2023).
2. Wesego, Daniel, and Pedram Rooshenas. "Multimodal ELBO with Diffusion Decoders." arXiv preprint arXiv:2408.16883 (2024).
3. Bounoua, Mustapha, Giulio Franzese, and Pietro Michiardi. "Multi-modal latent diffusion." Entropy 26, no. 4 (2024): 320.
4. Palumbo, Emanuele, Laura Manduchi, Sonia Laguna, Daphné Chopard, and Julia E. Vogt. "Deep generative clustering with multimodal diffusion variational autoencoders." In The Twelfth International Conference on Learning Representations. 2024.

**Questions:**

Please review the Weaknesses section and attempt to address the points raised there.

---

> ### Author Response · Authors · 2025-11-26
>
> Dear reviewer,
>
> Please checkout our updated paper as well as responses to all reviewers.
>
> 1. We have clarified that even the datasets in original submission are not toy datasets; they are used in multi-modal VAE papers published in the past year or two in top ML venues (see the references in our experiments section). Additionally we introduced CUB-HQ which has high resolution and more image background.
>
> 2. We have now conducted experimental comparisons with multimodal latent diffusion, in particular "Score-based multimodal autoencoders" on our datasets, ensuring the model architectures and latent dimensions are identical to our method. The results of this method, denoted SBM, are given in Table 1, Table 2, Table 9, and Table 10. In general SBM performs worse than our method, and the reason is intuitive: without resolving the correlation across views, VAEs learned individually for each view (as in stage 1 of SBM) will miss the shared information in challenging cases where the shared information is small relative to the entropy of input, which can not be recovered by diffusion on latent representations (stage 2 of SBM).
>
> 3. Novelty in using diffusion in latent space.
>
> - We emphasize that most of multi-modal latent diffusion papers do not couple latent diffusion and representation learning as tightly as we do: they tend to first learn individual VAEs for each modality, without resolving the correlation between different modalities, and use a second stage of diffusion modeling to couple the separately learned representations. In contrast, our latent prior is jointly trained with the rest of objective and regularization; the gradient of latent diffusion loss propagates to the encoders.
>
> - More importantly, none of these works perform disentanglement of the shared versus private variables. So even if we could evaluate their generative coherence, their methods do not offer the same level of control as we do, e.g., we can combine $z$ and $w$ from different data points or samples from the prior to generate new inputs.
>
> - We are not claiming to be the first one to use latent diffusion for multi-modal data, our novelty is instead the use of more flexible latent distribution to learn structured latent space (e.g., clusters) and disentangled representation.
>
> - We also would like to remind the reviewers that other terms in our final method are novel in the context of multimodal VAEs.
>
> 4. FIDs and CLIPScores are now provided whenever possible.
>
> 5. We did compare MMVAE+ with MoPoE, DMVAE in our experiments and MMVAE+ tends to perform the best, so it is natural for us to use it as the building block for our method. Our innovations on CrossMI and GenAug will improve on the other variants of multimodal VAEs.
> - To the best of our knowledge, our paper has compared with most important previous work in the area of multimodal VAEs and disentanglement (even the very recent work of DisentangledSSL) and the improvements are clear and hopefully convincing.

---

> > ### Comment · Reviewer_bE3u · 2025-11-26
> > **Reply**
> >
> > Dear Authors,
> >
> > Thank you for the reply. I'm still not convinced that your model is good enough. For example, look at Figures 5 and 13; even with a diffusion prior, the images generated are very bad. This may be because you are using a vanilla VAE, which has a lower quality in the first place, but still, I expected something better. [1,2] have better quality with diffusion prior / diffusion decoder compared to your result with FID values of less than 30. And I'm sure, the other will be better also when using diffusion prior and I doubt there could be some bug in your diffusion prior implementation.  I also gave the authors a better dataset of CelebAMask-HQ, which could be compared well across models in higher dimensions without using diffusion prior and across 3 modalities, but the authors didn't want to use it. For this reason, I will continue to maintain my current rating.
> >
> >
> > [1] Palumbo, Emanuele, Laura Manduchi, Sonia Laguna, Daphné Chopard, and Julia E. Vogt. "Deep generative clustering with multimodal diffusion variational autoencoders." In The Twelfth International Conference on Learning Representations. 2024.
> > [2] Wesego, Daniel, and Pedram Rooshenas. "Multimodal ELBO with Diffusion Decoders." arXiv preprint arXiv:2408.16883 (2024).

---

> > > ### Author Response · Authors · 2025-12-03
> > > **Additional experiments on CELEBAMASK-HQ**
> > >
> > > Dear reviewer,
> > >
> > > We just uploaded a new version which includes additional experiments on CELEBAMASK-HQ, in **Appendix-F**. We were able to use the code base of "Score-based multimodal autoencoders" to mostly reproduce the results for MMVAE+ and SBM-VAE. We kept the architecture to be the same for all methods. See Table 11 for the main results.
> > >
> > > Our method was tuned for the F-1 metric, for which we outperform baselines by a clear margin. Consistent with other experiments, each regularization term of our method contributes to better performance, and diffusion prior improves FIDs for image. For image generation, SBM-VAE does have better FIDs than ours (but our method still improves over reproduced MMVAE+). We think such results make sense, as our method spends capacity on extracting structured latent space enforced by regularizations.
> > >
> > > **We emphasize that multi-modal latent diffusion does not perform disentanglement of shared versus private variables**.
> > >
> > > Hopefully this clarifies your question on CELEBAMASK-HQ.

---

> ### Author Response · Authors · 2025-11-26
>
> Dear reviewer,
>
> 1. You are looking at the wrong figures: Figure 5 and 13 were for the old CUB experiments (low resolution images). Please see **Appendix E** for CUB-HQ experiments and high quality generations.
>
> 2. We have now high quality image dataset CUB-HQ and we already have a 5-view dataset PolyMNIST. We do not see significant benefit in using another dataset for demonstration purpose.
>
> - Note we are already using **more challenging** versions of PolyMNIST (64x64, significant private information in quadrants) and CUB-HQ than prior work such as MMVAE+ and [1] (according to the MMVAE+ code base, the authors were using 64x64 images for representation learning, with birds cropped using bounding boxes; this lower resolution version was what we used in Sec 4.2). The diffusion model of [1] is trained after an initial MMVAE+ stage (and another discrete latent variable learning stage), similar to our DiT denoiser for CUB-HQ. The differences in FID can potentially also come from image resolution (e.g., 256x256 vs 512x512 inputs) and particular diffusion model architecture, though we are using the same diffusion denoiser implementation for all methods in Appendix E, so we could see the relative merit of methods.
>
> [1] Palumbo, Emanuele, Laura Manduchi, Sonia Laguna, Daphné Chopard, and Julia E. Vogt. "Deep generative clustering with multimodal diffusion variational autoencoders." In The Twelfth International Conference on Learning Representations. 2024.
>
> - We additionally demonstrated different methods on the TCGA multi-omics data as used by Lee & van der Schaar, AISTATS 2021 and Liu et al, ICML 2023; similar dataset was used in DisentangledSSL. **Generation and FID are not the only purposes of multi-modal representation learning**.
>
> -- Changhee Lee and Mihaela van der Schaar. A variational information bottleneck approach to multi-omics data integration. In AISTATS, 2021.
>
> -- Tennison Liu, Jeroen Berrevoets, Zhaozhi Qian, and Mihaela Van Der Schaar. Learning representations without compositional assumptions. In ICML, 2023
>
> - Most importantly, we think you missed the key point that our focus is **disentangled** low dimensional latent representation. Despite that multi-modal latent diffusion can generate, **their methods do not offer the same level of control as we do**, e.g., we can combine z and w from different data points or samples from the prior to generate new inputs. **We also could not evaluate the information content of latent spaces of Score-based Multimodal Autoencoders as two partitions (shared vs private)**.
>
> 3. We have uploaded as supplementary materials our code for running Score-based Multimodal Autoencoders on our PolyMNIST-Quadrant datasets (including architecture, trainer files for both steps; other files are the same as original repo). We welcome you to check if there is bug in our use of the method and we are happy to rerun it.

---

### Official Review · Reviewer_1dPy · 2025-11-01

**Soundness:** 3
**Presentation:** 3
**Contribution:** 3
**Rating:** 6
**Confidence:** 5

**Summary:**

The paper “Disentanglement of Variations with Multi-Modal Generative Modeling (IDMVAE)” proposes a new multimodal variational autoencoder designed to cleanly separate shared and private sources of variation across data modalities. Standard multimodal VAEs, such as MMVAE and MoPoE, often suffer from latent leakage, where modality-specific information contaminates the shared latent space, causing incoherent cross-modal generation and poor disentanglement. IDMVAE introduces three complementary mechanisms to address this.

1.	A cross-view mutual information maximization term aligns shared representations by encouraging different modality-specific encoders to capture the same underlying latent factors.
2.	A generative augmentation cycle removes redundancy between shared and private latents by decoding “mixed” latent pairs into synthetic samples and enforcing that shared latents remain invariant while private latents retain modality-specific structure.
3.	A latent diffusion prior replaces the standard Gaussian prior with a learned diffusion model that can represent complex, multimodal latent distributions, reducing prior–posterior mismatch and improving unconditional generation quality.

The resulting objective extends the MMVAE+ ELBO with information-theoretic and generative-consistency regularizers, all trainable end-to-end. Experiments on PolyMNIST-Quadrant, CUB (image–text), and TCGA (multi-omics) show that IDMVAE achieves substantially cleaner separation of shared vs. private factors, higher conditional and unconditional coherence, and improved latent alignment compared to strong baselines. Ablation studies demonstrate that cross-view MI alignment improves cross-modal reconstruction, the generative augmentation reduces leakage, and the diffusion prior enhances sample quality.

**Strengths:**

Originality:\
Introduces a principled formulation of multimodal disentanglement by combining cross-view mutual-information maximization with generative augmentation, directly targeting the separation of shared and private factors.\
Conceptually unifies ideas from contrastive learning, information bottleneck, and generative modeling in a single probabilistic framework.

Quality:\
Methodologically robust: each loss term is clearly motivated by known weaknesses of multimodal VAEs (e.g., posterior collapse, shared–private leakage).\
The generative augmentation mechanism leverages the model’s own decoder for cycle consistency, avoiding handcrafted data augmentations\
Comprehensive experiments across synthetic (PolyMNIST-Quadrant) and real (CUB, TCGA) datasets demonstrate consistent improvements in disentanglement, coherence, and generation quality.\
Ablation studies isolate the contribution of each component (CrossMI, GenAug, diffusion prior) and show interpretable, consistent improvements.\

Clarity:\
The paper is well-structured and clearly written, with each component (CrossMI, GenAug, diffusion prior) well motivated and described.\
Mathematical notation and objective formulation are concise and consistent.

Significance:\
Addresses a challenge in multimodal learning of separating  shared vs. private information with a practical, modular solution applicable to existing architectures.\
Demonstrates that learned latent priors (diffusion models) can substantially improve multimodal VAEs, opening a new direction for hybrid diffusion–VAE research. However, it lacks a clear comparison with other multi-modal (latent) diffusion models, such as [1,2].

Refs:
[1] Bounoua, Mustapha, Giulio Franzese, and Pietro Michiardi. "Multi-modal latent diffusion." Entropy 26.4 (2024): 320. \
[2] Chen, Changyou, et al. "Diffusion models for multi-modal generative modeling." arXiv preprint arXiv:2407.17571 (2024).

**Weaknesses:**

The connection between maximizing $I(z_m; z_n)$ and achieving conditional sufficiency $I(x_m;x_n|z)=0$ remains heuristic. Under what assumptions (e.g., conditional independence or additive latent structure) do the suggested losses approximate or minimal-sufficiency conditions.\
The InfoNCE estimator can be sensitive to negative-sample size, temperature, and batch composition. The paper fixes these hyperparameters without ablation or analysis.\
Adding contrastive, augmentation, and diffusion losses likely increases computational cost. The paper lacks a discussion of these computation costs.\
You compare to MMVAE variants but not to contrastive approaches such as [3-4], or as mentioned above, latent diffusion models [1-2]. \
The generative augmentation term is described as reducing redundancy between z and w_m, but the paper does not analyze how effectively this enforces $I(z; w_m) \to 0$ in the experiments.



Refs: \
[3] Liang, Paul Pu, et al. "Factorized contrastive learning: Going beyond multi-view redundancy." Advances in Neural Information Processing Systems 36 (2023): 32971-32998.\
[4] Dufumier, Benoit, et al. "What to align in multimodal contrastive learning?." arXiv preprint arXiv:2409.07402 (2024).

**Questions:**

1.	Is the diffusion prior applied only to the shared latent z, or also to private w_m? If only to z, why is a simple Gaussian sufficient for the private components?
2.	How do the MI and cycle constraints correspond to orthogonality or covariance restrictions in linear Bayesian joint additive factor models such as [5]?
3.	The paper does not mention a β-VAE coefficient. Is β fixed at 1, and do $\lambda_{\text{MI}}$, $\lambda_{\text{GenAug}}$ effectively play this role? How do these hyperparameters influence the average posterior entropies $H[q(z|x_m)] $ and $H[q(w_m|x_m)] $ over training.
4.	Some VAE models e.g. [6] address the inference-consistency problem in multimodal VAEs, i.e. the inability to obtain coherent subset posteriors such as $p(w_2 \mid x_1)$ via a tighter ELBO and flexible encoder aggregation. Does IDMVAE provide an alternative route to this goal by enforcing independence between $w_m$ and the shared latent z, thereby avoiding aggregation bias or looser bounds for modalities with high conditional mutual information?
5.	The CrossMI loss is motivated by encouraging $I(z_m;x_n)\approx I(x_m;x_n)$, implying that the shared latent fully explains cross-modal dependence. Would this not render private latents redundant? How is this balanced to prevent collapse of private variation, and does this depend on dataset redundancy?
6.	Integrating a diffusion prior end-to-end into a multimodal VAE can introduce instability, as seen in recent works like REPA-E [7] or uses some annealing tricks [8]. The paper does not discuss training schedules, warm-up strategies, or whether gradients from the diffusion loss are stopped for encoder updates.

Refs:
[5] Anceschi, Niccolo, et al. "Bayesian joint additive factor models for multiview learning." arXiv preprint arXiv:2406.00778(2024).\
[6] Hirt, Marcel, et al. "Learning multi-modal generative models with permutation-invariant encoders and tighter variational objectives." Transactions on Machine Learning Research.\
[7] Leng, Xingjian, et al. "Repa-e: Unlocking vae for end-to-end tuning with latent diffusion transformers." arXiv preprint arXiv:2504.10483 (2025).\
[8] Vahdat, Arash, Karsten Kreis, and Jan Kautz. "Score-based generative modeling in latent space." Advances in neural information processing systems 34 (2021): 11287-11302.

---

> ### Author Response · Authors · 2025-11-26
>
> We thank the reviewer for checking the previous updated version. We have been busy implementing and comparing with the multimodal latent diffusion method, and we just posted a new version to address your concerns.
>
> ## 1. Comparison with multimodal latent diffusion approach.
>
> We have now conducted experiments with "Score-based multimodal autoencoders" on our datasets, ensuring the model architectures and latent dimensions are identical to our method. The results of this method, denoted SBM, are given in Table 1, Table 2, Table 9, and Table 10. In general SBM performs worse than our method, and the reason is intuitive: without resolving the correlation across views, VAEs learned individually for each view (as in stage 1 of SBM) will miss the shared information in challenging cases where the shared information is small relative to the entropy of input, which can not be recovered by diffusion on latent representations (stage 2 of SBM).
>
> --  Wesego, Daniel, and Amirmohammad Rooshenas. "Score-based multimodal autoencoders." arXiv:2305.15708 (2023).
>
> ## 2. Theoretical justification of conditional sufficiency.
>
> - Our goal is to achieve sufficiency for $z_1$, i.e., $I(z_1;x_2)=I(x_1;x_2)$ where $z_1$ is a representation of $x_1$. If this is achieved, by using the chain rule twice, we have
> \begin{align*}
> I(x_1;x_2|z_1) =I(x_1,z_1;x_2) - I(z_1;x_2) =I(x_1;x_2) + I(z_1;x_2|x_1) - I(z_1;x_2) = I(z_1;x_2|x_1) =0
> \end{align*}
> where the last equality is due to all variability of $z_1$ being captured by $x_1$ and therefore $z_1$ is conditionally independent of $x_2$. As a result, conditioned on a sufficient $z_1$, the input views become independent. This argument could be found in (Federici et al., 2020) which we cited. Additionally, we argued in Section 2.2 (line 120 to line 125) that the sufficiency is hard to achieve with likelihood modeling alone, due to the presence of $w$, and therefore we resort to contrastive learning based lower bound.
>
> - In section 2.3, we argued that $w$ can be recovered by essentially having $I(z,w_m;x_m)=H(x_m)$ while $I(z;w_m)=0$. Private latent will not be redundant but has to explain variations of the input not captured by $z$. Our assumptions are given in Figure 1: shared and private variables jointly generate input, and they are independent from each other. We can make this more explicit in the future version.
>
> - To understand how effective is our $\mathcal{L}_{\text{GenAug}}$ in reducing redundancy, we measure empirical mutual information between $z$ and $w_m$, using the estimate of $\beta$-TCVAE for total correlation on our validation set:
> | Method |MI estimate |
> |:-----|:---:|
> | MMVAE+ | 8.3 |
> | IDMVAE with  $\lambda_1$=0 | 1.9 |
> | IDMVAE with $\lambda_2$=0 | 7.4 |
> | IDMVAE | 7.4 |
> | IDMVAE + diffusion prior| 7.7 |
>
> You can see that if we only use Generative Augmentation (i.e., with $\lambda_1$=0, Cross-MI is off ), we can drive the MI estimate to be the lowest value 1.9 (but of course in this case we fail to capture the shared information).
>
> Interestingly, in the early stage of our work we tried to use the  $\beta$-TCVAE implementation of total correlation for removing redundancy between $(z,w)$ but it did not have a positive effect; so our  Generative Augmentation approach may have other inductive bias that helps achieve disentanglement.
>
>
> ## 3. Computational cost and training stability.
>
> We thank reviewers for questions on the details, we will open-source our code when the paper is accepted.
>
> - We did not tune temperature in contrastive loss; it is set to 1.0. We did make sure we have a large enough batch size (e.g., 128).
>
> - We thank reviewer for references [5,6]; they potentially can improve the MMVAE+ loss.
>
> - We did not heavily tune $\beta$ in the VAE objective: in the end we use $\beta=2.5$ for PolyMNIST which was shown to achieve a sweet spot on generation quality and coherence by MMVAE+, we use $\beta=1.0$ for CUB; tuning them did not make huge difference on our qualitative measures. We note that our CrossMI and GenAug losses learn very structured representation that are not achievable by tuning $\beta$.
>
> - Orthogonal loss for encouraging uncorrelatedness between $z$ and $w$ was actually used by DisentangledSSL which we compared with.
>
> - Regarding the training stability with diffusion prior, we thank the reviewers' suggestion of [7,8]. We use diffusion priors for both $z$ and $w$ without stopping gradient. Indeed, if we set the coefficient of diffusion prior to be too large, training is stuck since the beginning. Instead of using a schedule, we simply set the diffusion loss weight to be smaller like 0.1 in our implementation. Please check out the 2D visualization of latents in Figure 9,10,21,22, where samples from the diffusion prior well match the VAE posteriors.
>
> - Overall, we believe a good algorithm should not require much tuning, so our focus was on the loss coefficients of CrossMI and GenAug. Your suggestion on the tunable parameters may further improve our results.

---

> > ### Author Response · Authors · 2025-11-26
> >
> > Dear reviewer 1dPy,
> >
> > We thank you for your Post-rebuttal update based on the previous version. Please take a look at the new version we posted today and let us know if we cleared all your previous questions.

---

### Author Response · Authors · 2025-11-21
**Response to all reviewers**

We thank the reviewers for constructive feedback. We will post individual responses to each reviewer soon. Here we address the common concerns.

1. Datasets and High quality generation

The dataset we used are commonly found in Multi-modal VAE papers published in top ML venues in just the past 2-3 years, as shown by references in the main paper.

Now we would like the reviewers to check out our new experiments on **high-resolution CUB (CUB-HQ)**, in **Appendix E** of the updated paper (we did not directly modify the main paper to avoid significant re-formatting). We will replace the previous CUB experiments with CUB-HQ experiments in the final version. In CUB-HQ, we use the original CUB images without cropping out the bird portion using bounding boxes, and use a **high 256x256 resolution**. This means the high-resolution images contain richer backgrounds, and CUB-HQ is therefore more challenging for representation learning. We preprocess the dataset using pretrained VAE as in DiT, and train IDMVAE with relatively small resnet architecture. We then train DiT denoisers on top of IDMVAE samples, to produce high resolution images, following [1] and [2].

The results show that our method outperforms baselines by a clear margin, and each regularization term contributes. We provide not only latent classification (Table 9) but also **FID and CLIPScore** (Table 10) as requested by reviewers to assess the quality of generation (samples given in Figures 16--19). Again our method beats the baselines and all loss terms lead to improvement. We hope these results clarify reviews’ concerns on scaling up to high resolution generation: with careful use of pretrained models, IDMVAE can efficiently learn disentangled representations that lead to controllable generation of complex modalities.

2. Related works on diffusion for multi-modal data

We thank the reviewers for bringing up papers on latent diffusion for multimodal learning, most importantly [3,4].

- We emphasize that most of these references do not couple latent diffusion and representation learning as tightly as we do: they tend to first learn individual VAEs for each modality, without resolving the correlation between different modalities, and use a second stage of diffusion modeling to couple the separately learned representations. In contrast, our latent prior is jointly trained with the rest of objective and regularization; the gradient of latent diffusion loss propagates to the encoders.

- More importantly, **none of these works perform disentanglement of the shared versus private variables**. So even if we could evaluate their generative coherence, their methods do not offer the same level of control as we do, e.g., we can combine $z$ and $w$ from different data points or samples from the prior to generate new inputs.

- We are not claiming to be the first one to use latent diffusion for multi-modal data, our novelty is instead the use of more flexible latent distribution to learn structured latent space (e.g., clusters) and *disentangled* representation. Nonetheless, we will cite these papers and add discussions in a future version.

- Our application of DiT denoisers in CUB-HQ is similar to the method described in [2,5]; their works previously demonstrated that conditional diffusion yields high-quality results for multi-modal data, which we acknowledged in the updated version.

All in all, the existence of prior work on diffusion models for multi-modal data does not undermine our novelty in multi-modal disentanglement.

- **We have now conducted experimental comparisons with multimodal latent diffusion**, in particular [4] on our datasets, ensuring the model architectures and latent dimensions are identical to our method. The results of this method, denoted SBM, are given in Table 1, Table 2, Table 9, and Table 10. In general SBM performs worse than our method, and the reason is intuitive: without resolving the correlation across views, **VAEs learned individually for each view (as in stage 1 of SBM) will miss the shared information** in challenging cases where the shared information is small relative to the entropy of input, which can not be recovered by diffusion on latent representations (stage 2 of SBM).


[1] Pandey et al. DiffuseVAE: Efficient, controllable and high-fidelity generation from low-dimensional latents. TMLR, 2022.

[2] Palumbo et al. Deep generative clustering with multimodal diffusion variational autoencoders. In ICLR, 2024.

[3] Bounoua et al. "Multi-modal latent diffusion." Entropy 26.4 (2024): 320.

[4] Wesego, Daniel, and Amirmohammad Rooshenas. "Score-based multimodal autoencoders." arXiv preprint arXiv:2305.15708 (2023).

[5] Wesego, Daniel, and Pedram Rooshenas. "Multimodal ELBO with Diffusion Decoders." arXiv preprint arXiv:2408.16883 (2024).

---

### Meta-Review · Area_Chair_kgG9 · 2026-01-08

**Summary:**

This work has important strength raised by the reviewers, but several weaknesses. Its scoring was 6, 2, 4.

Main concerns regard the clarification of some punctual parts of the method, missing discussion about hyper/parameters' setting and sensitivity, and some critics about the use of heavy heuristics in some parts of the proposed method. Also, no discussion about computational cost.

Other comments address the use of only toy datasets, questionable novelty, quoting of a limited number of SotA baselines, incomplete performance analysis (FID not used), and missing ablations. Discussion was engaged twice, but authors seem to reply convincingly.

Other issues are again on novelty, training efficiency and runtime analysis, and, in the validation phase, limited comparisons with SotA methods, insufficient visualization of generated samples, and the need to address large scale dataset.

**Reviewer Concerns:**

All raised remarks have been addressed by authors, and in a satisfactory way. Comments are several but they seem not so critical not to be satisfactorily addressed in a rebuttal/revision phase.

The most critical reviewer engaged in a discussion, but authors are convincing in their reply.

**Reviewer Scores:**

Unclear which reviewer can raise the score, unlikely Rev. bE3u (who scored 2), more likely Rev. wgHU.

---

### Decision · Program_Chairs · 2026-01-26

Accept (Poster)